# Language Generation with Feedback: Queries and Mistakes

**Steve Hanneke** [1]  **Amin Karbasi** [2]  **Anay Mehrotra** [3]  **Grigoris Velegkas** [4]

## Abstract

We investigate language generation in the limit (Kleinberg & Mullainathan, 2024; Li et al., 2025) in variants where the generator receives some feedback based on its "actions." We study two such variants. In the first, which is inspired by Littlestone's model of online learning, the generator observes whether it made a mistake at each iteration. In the second, introduced by Charikar & Pabbaraju (2025), the generator can query whether a string belongs to the target language.

Our main result is a characterization of collections that are generable with mistake feedback. Using similar techniques, we also characterize when generation is possible in the query model with set-based generators; set-based generators have been studied in several works (Charikar & Pabbaraju, 2025; Kalavasis et al., 2025; Kleinberg & Wei, 2025; Li et al., 2025). Beyond the characterizations themselves, we derive several implications. First, our results imply new closure properties for generation with mistake and query feedback. Second, our results show that, under feedback, generation is robust to noise: it remains possible with arbitrary contamination in the adversary's examples and with finite contamination in the feedback. Third, our techniques also yield new sufficient and necessary conditions for generation *without* feedback among other implications.

## 1. Introduction

How do humans or Large Language Models (LLMs) learn to generate meaningful language? Perhaps the most basic description of the task is to, given a corpus of text, generate meaningful or *valid* sentences that are not present in the corpus. Despite having a simple description and being a central conceptual problem in learning theory, the mechanisms of language generation have historically resisted a clean theoretical explanation.

Toward this end, Kleinberg & Mullainathan (2024) recently proposed a rigorous framework for this problem, in the spirit of classical online learning and identification models (Gold, 1967; Littlestone, 1988). The process begins with an adversary selecting a target language $K$ from a collection of languages $\mathcal{L}$ and fixing an enumeration of that language.[1] At each step $n \geq 1$, the adversary reveals the $n$-th element, $x_n$ of the enumeration. Having observed the ordered history $H_n = (x_1, \ldots, x_n)$, and hence the seen set $S_n = \{x_1, \ldots, x_n\}$, the generator must produce a new output $w_n \notin S_n$ intended to be a valid, unseen string in $K$.

A generator $\mathcal{G}$ is said to be successful if it learns to "generate from collection $\mathcal{L}$ in the limit." Formally, this means that for any language $K$ in collection $\mathcal{L}$ and any enumeration, there exists a finite round $n^\star$ after which the generator never fails. In other words, for all rounds $n \geq n^\star$, the produced string $w_n$ must be a valid member of the target language that has not yet appeared in the history, *i.e.*, $w_n \in K \setminus S_n$.

This framework is rooted in the seminal work of Gold (1967) on language *identification*, which requires the learner to identify the target $K$ (*i.e.*, learn it exactly). While identification is impossible for most non-trivial language classes, Kleinberg & Mullainathan (2024) demonstrated that shifting the objective to *generation* makes the task feasible for a broad class of language collections—including any countable collection of languages. This surprising finding has catalyzed a wave of recent research (*e.g.*, (Li et al., 2025; Kalavasis et al., 2025; Charikar & Pabbaraju, 2025; Raman & Raman, 2025)); see Section 1.3.

Despite this remarkable positive result, several aspects of the current landscape of generation remain unsatisfactory when considering *uncountable* collections of languages. For example, Hanneke et al. (2025); Bai et al. (2026) identified a particularly counterintuitive property: generability is not preserved under (finite) unions. That is, there ex-

---

[1]Purdue University [2]Cisco Foundation AI [3]Stanford University [4]Google Research. Correspondence to: Anay Mehrotra <anaymehrotra1@gmail.com>, Grigoris Velegkas <gvelegkas@google.com>.

*Proceedings of the $43^{rd}$ International Conference on Machine Learning*, Seoul, South Korea. PMLR 306, 2026. Copyright 2026 by the author(s).

---

[1]Formally, an enumeration of $K$ is an infinite sequence $x_1, x_2, \ldots$ (potentially with duplicates) such that $x_i \in K$ for all $i$ and every $x \in K$ appears at some index in the sequence.

ist collections $\mathcal{L}_1, \mathcal{L}_2$ that are individually generable, yet their union $\mathcal{L}_1 \cup \mathcal{L}_2$ is not. This stands in sharp contrast to most tasks in learning theory, such as binary classification and online learning, where finite unions do not exhibit this pathology. Moreover, Bai et al. (2026) showed that generability is brittle to contamination, in the form of either incorrect examples in the stream or omitted elements of the target language from the stream: there are collections that are generable in the limit, yet if the adversary enumerates even a single incorrect element, or omits a single element from the target, the collection is no longer generable.

A potential reason for these limitations is that Kleinberg & Mullainathan (2024)'s framework does not incorporate *feedback*, despite the fact that both humans and LLMs improve significantly when feedback is provided. In this work, we aim to bridge this gap by exploring two types of feedback: **i)** feedback regarding past mistakes made by the learner, and **ii)** membership queries, which allow the generator to ask whether a specific string belongs to the target language $K$ (Charikar & Pabbaraju, 2025; Bai et al., 2026).

### 1.1. Our Results and Their Implications

Our main results are characterizations of which language collections are generable under the above feedback models for different notions of generation. A unifying theme in our characterizations is a new notion of an inner-cover of a collection $\mathcal{L}$ that we introduce.

**Inner-covers.** Informally, a collection $\mathcal{C}$ is an inner-cover of another collection $\mathcal{L}$ if for every language $L \in \mathcal{L}$ there exists some $C \in \mathcal{C}$ with $C \subseteq L$ and $|C| = \infty$; see Definition 4. Throughout, we will be especially interested in inner-covers that have countably many elements.

**Element- and Set-Based Generators.** We study two types of generators: element-based and set-based. Element-based generators work as described above: in each round, they output a single string $w_n$, and success means that after some finite number of rounds the generator outputs only valid unseen strings, *i.e.*, $w_n \in K \setminus S_n$. Set-based (or autoregressive) generators impose a stronger requirement: in each round they output an *infinite* set $W_n$ of strings, and success means that after some finite number of rounds all of these strings are valid and unseen, *i.e.*, $W_n \subseteq K \setminus S_n$. Because set-based generation demands infinitely many valid and unseen strings each round, it is a more demanding form of generation. This has motivated several recent works to study this notion (Kalavasis et al., 2025; Charikar & Pabbaraju, 2025; Kleinberg & Wei, 2025; Li et al., 2025).

**Characterizations for feedback models.** Our first result characterizes generation with mistake feedback, where after each round the generator learns if its output was in $K$. If the generator is set-based and outputs set $W_n$, then it sees if the "first element" of $W_n$ is in $K$; so set-based generators get a

similar amount of feedback as element-based generators.

**Informal Theorem 1.1** (Main characterizations for mistake feedback; see Theorems 3.1 and 3.2)**.** *A collection $\mathcal{L}$ is generable with mistake feedback if and only if it admits an inner-cover of countable size. Moreover, $\mathcal{L}$ is generable by an element-based generator if and only if it is generable by a set-based generator.*

Next, we study query feedback, where in each round the learner may ask whether a string of its choice belongs to the target language $K$. Here, element-based and set-based generation no longer coincide; nevertheless, set-based generation admits the same countable-inner-cover characterization.

**Informal Theorem 1.2** (Main characterizations for query feedback; see Theorems 3.3 and 3.4)**.** *There is a collection $\mathcal{L}'$ that is generable with query feedback but not set-generable with query feedback. Further, a collection $\mathcal{L}$ is set-generable with query feedback if and only if it admits a countable inner-cover.*

The above results have several consequences, sometimes combined with prior techniques, which we outline below.

**Closure properties.** The countable-inner-cover characterization immediately yields several new closure properties. For example, set-based generation with query feedback and both modes of generation with mistake feedback are closed under countable unions: if collections $\{\mathcal{L}_1, \mathcal{L}_2, \dots\}$ are generable in the appropriate model, then $\bigcup_{i \in \mathbb{N}} \mathcal{L}_i$ is also generable in the same model (Corollaries A.3 and 3.6). This extends the union-closedness result of Bai et al. (2026) in the query model from element-based generators to set-based generators. The same perspective also yields closure under other operations, such as Cartesian products, by combining inner-covers in the natural way (see Section 3.4.1).

**Generation with Zero Examples.** We show that under feedback the adversary's examples are unnecessary: with feedback, the generator can succeed even in a "zero-example" setting where the stream provides no informative samples and only the feedback channel remains (Corollary 3.7).

**Noisy generation with feedback.** Combining our generators with the finite-expansion subroutine of Mehrotra et al. (2025), we show that feedback-based generation is robust to finite contamination. In particular, the learner tolerates *arbitrary* contamination in the example stream and finitely many corruptions in the feedback (Corollary 3.8). This stands in contrast to the no-feedback model, which can fail under even a single corrupted example (Bai et al., 2026).

**Implications for generation without feedback.** Characterizing generation in the limit *without* feedback is a major open problem in this line of work on language generation in the limit. Somewhat surprisingly, we show that our notion of inner-covers also has implications for this problem. Infor-

mally, we show that generability of a collection $\mathcal{L}$ (without feedback) is "sandwiched" between having inner-covers of finite and countable size:

1. If $\mathcal{L}$ has a finite inner-cover then it is generable;
2. If $\mathcal{L}$ has no countable inner-cover then it is not generable;
3. There is a non-generable $\mathcal{L}$ with a countable inner-cover;
4. There is a generable $\mathcal{L}$ with a countable inner-cover.

Moreover, perhaps surprisingly, we show that generation is achievable if and only if the seemingly stronger requirement of set-based generation is achievable. Lastly, we establish a technical property that could be useful for eventual characterizations of generability: if $\mathcal{L}$ is generable by generator $\mathcal{G}$, then for each $L \in \mathcal{L}$, there is a finite locking history over $L$ such that once this history appears as a prefix, $\mathcal{G}$ always generates correctly on valid continuations.

## 1.2. Technical Overview

Next, we overview some of the techniques we develop. Instead of focusing on a specific result, we present tools that connect element-based generation, set-based generation, and countable inner-covers—tools we use repeatedly throughout our proofs. For simplicity, we present these tools in the model *without* any feedback. In our proofs, we need to handle feedback, which introduces additional challenges.

**Element-Based Generator to Set-based Generator.** We show that element-based generators can often be *converted* to set-based generators using a new "simulation" argument. Our argument rests on the following property:

> **(Generation Locking Histories; Lemmas A.1 and A.8)** If $\mathcal{L}$ is generable, then for every successful sequence-input generator $\mathcal{G}$ and every $L \in \mathcal{L}$ there exists some *finite* locking history $\sigma_L$, such that, after $\sigma_L$ appears as a prefix, $\mathcal{G}$ outputs a fresh element of $L$ at every subsequent timestep for every valid continuation.

To establish this, we rely on a diagonalization argument, which shows that if $\mathcal{L}$ does not have this property, then the adversary can always find an extension of the current enumeration so that the generator makes a mistake.

Given such a locking history, we can convert an element-based generator $\mathcal{G}$ to a set-based generator $\mathcal{G}_S$ by self-simulation. At step $t$, given ordered history $H_t$, recursively append the generator's own previous outputs to the history and output the infinite set of strings produced along this simulated continuation. In words, the set-based generator $\mathcal{G}_S$ outputs the infinite sequence of examples obtained by feeding outputs of $\mathcal{G}$ to its input stream in a fashion resembling auto-regressive generation. We use the above result to conclude that, once the locking history has appeared, all

elements of this infinite set are valid and unseen elements of $L$.

**Set-based Generation to Countable Inner-Cover.** We also establish a connection between set-based generators and countable inner-covers. Here, we show that the number of distinct outputs of any (valid) set-based generator is at most countably many. We then use these outputs to construct a countable inner-cover of $\mathcal{L}$.

**Generating via Countable Inner-Covers.** We also connect countable inner-covers back to generation. Since Kleinberg & Mullainathan (2024) (and follow-up works) have designed several generators for countable collections, a natural idea is to simply run their generator on the cover $\mathcal{C}$ and hope it guarantees generation for $\mathcal{L}$. Unfortunately, there is a subtle issue: the inner-cover $\mathcal{C}$ merely guarantees that for any $K \in \mathcal{L}$ there exists some $C \in \mathcal{C}$ that is a *subset* of $K$, but $K$ itself may not belong to $\mathcal{C}$. Because of this, the given enumeration effectively appears as if it contains (perhaps infinitely many) noisy examples, which makes generation impossible (Bai et al., 2026; Mehrotra et al., 2025). Fortunately, we are able to bypass this using feedback.

**Contamination Robustness.** Lastly, to obtain the results for generation despite contamination in the provided feedback, we show how to combine our ideas on inner-covers with the "finite expansion" subroutine of Mehrotra et al. (2025) to handle contaminated inputs.

## 1.3. Related Work

Our work builds on the language generation in the limit framework of Kleinberg & Mullainathan (2024), as well as the model of feedback in language generation introduced by Charikar & Pabbaraju (2025) and further studied by Bai et al. (2026). Recently, there has been a flourishing line of work studying different perspectives of language generation in the limit (*e.g.*, (Li et al., 2025; Kalavasis et al., 2025; Charikar & Pabbaraju, 2025; Raman & Raman, 2025; Peale et al., 2025; Kleinberg & Wei, 2025; 2026; Hanneke et al., 2025; Mehrotra et al., 2025; Charikar et al., 2025; Karbasi et al., 2025; Arenas et al., 2025; Anastasopoulos et al., 2026)). Interestingly, a concept related to the inner-covers we use in our work has appeared in the work of Braverman et al. (2021), in a very different learning-theoretic setting. Below, we discuss prior works that are most relevant to our setting.

**Feedback in Language Generation.** Charikar & Pabbaraju (2025) introduced a model of language generation with feedback, where in every timestep $t \in \mathbb{N}$ the learner is allowed to ask whether a string $w$ of its choice belongs to the target language $K$. Importantly, if the answer is affirmative, the learner is allowed to output the queried string. The authors showed a separation between uniform generation with and without feedback. Subsequently, Bai et al. (2026) studied the role of feedback in uncountable collections of

languages; their results showed that while a *finite*[2] amount of feedback does not change which collections are generable in the limit, allowing for feedback in each round of the interaction strictly increases the set of generable collections.

**Language Generation with Noise.** Recent works have also explored language generation with contaminated inputs, in the form of noisy examples or omitted examples. Raman & Raman (2025) investigated language generation in a model where an adversary can introduce *finite* amount of incorrect examples in the stream, focusing on countable language collections. Bai et al. (2026) continued this investigation to uncountable collections and also studied a model where the adversary can omit *infinitely* many elements of $K$. Mehrotra et al. (2025) studied a setting of generation for countable collection of languages that allows for infinite noisy examples in the input as well as infinite omissions. Kleinberg & Wei (2026) studied a model of *partial* identification of the target given a stream with no noise but infinite omissions.

## 2. Model and Preliminaries

We now present the necessary background on language generation in the limit.

**Notation.** Let $\Sigma$ be a finite alphabet and let $\Sigma^*$ denote the set of all finite strings over $\Sigma$. We use $\mathcal{X}$ to denote an arbitrary countable universe of strings; typically $\mathcal{X} = \Sigma^*$. We fix an arbitrary canonical ordering of the domain $\mathcal{X}$ and denote it by $(\overline{x}_1, \overline{x}_2, \dots)$. A *language* is an infinite subset $L \subseteq \mathcal{X}$. A collection of languages is a set $\mathcal{L} \subseteq 2^{\mathcal{X}}$ (which may be countable or uncountable). When $\mathcal{L}$ is countable, we fix an explicit enumeration $\mathcal{L} = \{L_1, L_2, \dots\}$.

**Generating algorithms.** For a finite sequence $h = (h_1, \dots, h_n) \in \mathcal{X}^n$, write

$$\mathrm{supp}(h) \coloneqq \{h_1, \dots, h_n\}$$

for its underlying set of observed strings. A *generating algorithm* (or simply generator) is a sequence $\mathcal{G} = (\mathcal{G}_n)_{n \in \mathbb{N}}$ where each $\mathcal{G}_n : \mathcal{X}^n \to 2^{\mathcal{X}}$ maps a length-$n$ *ordered* history of observed examples to a (possibly infinite) set of candidate outputs.[3] If $\mathcal{G}$ always outputs singletons, we say it is an *element-based* generator. If $\mathcal{G}$ outputs *infinite* sets, we say it is a *set-based* generator. Notice that set-based generation resembles the auto-regressive generation of modern LLMs: after seeing sufficiently many valid sentences, they can generate a vast amount of new valid sentences.

**Enumeration of languages.** An *enumeration* of a language

---

[2] In this model, the learner is only allowed to submit a finite number of queries throughout the interaction.

[3] The original formulation of Kleinberg & Mullainathan (2024) outputs a single string each round; later work (Kleinberg & Wei, 2025; Li et al., 2025; Kalavasis et al., 2026; Charikar & Pabbaraju, 2025) allows outputting a set (or, equivalently, a sampling procedure) once sufficient training data have been observed.

$K$ is an infinite sequence $x_1, x_2, \dots$ (possibly with repeats) such that each $x_t \in K$ and every element of $K$ appears at some time. We write

$$H_n \coloneqq (x_1, \dots, x_n), S_n \coloneqq \mathrm{set}(H_n) = \{x_1, \dots, x_n\}$$

for, respectively, the length-$n$ ordered history and the underlying set of the first $n$ elements in the adversary's enumeration.

### 2.1. Language Generation in the Limit

We now formally define language generation in the limit.

**Definition 1** (Language Generation in the Limit (Kleinberg & Mullainathan, 2024)). *Fix some $K$ from the language collection $\mathcal{L}$ and a generating algorithm $\mathcal{G} = (\mathcal{G}_n)$. At each step $n$, let $H_n = (x_1, \dots, x_n) \in K^n$ be the ordered history seen so far and let $S_n = \mathrm{set}(H_n)$ be its underlying set. The algorithm $\mathcal{G}$ is said to generate from $K$ in the limit if, for all enumerations of $K$, there is some $n^* \in \mathbb{N}$ such that for all steps $n \geq n^*$,*

1. *if $\mathcal{G}$ is element-based, then $\mathcal{G}_n(H_n) \in K \setminus S_n$;*
2. *if $\mathcal{G}$ is set-based, then $\mathcal{G}_n(H_n) \subseteq K \setminus S_n$.*

*The collection $\mathcal{L}$ allows for generation in the limit (or is generable) if there is an algorithm $\mathcal{G}$ that generates from $K$ in the limit for any $K \in \mathcal{L}$.*

To gain some intuition, consider the following example.

**Example 2.1** (Length-threshold languages). Fix a finite alphabet $\Sigma$ and consider the countable collection of length-threshold languages $\mathcal{L} = \{L_1, L_2, \dots\}$ where, for each $\ell$, $L_\ell \coloneqq \{x \in \Sigma^* : |x| \geq \ell\}$. Suppose the target is $K = L_{\ell^\star}$ and the adversary selects the enumeration $x_1, x_2, \dots$. After observing $S_n$, we know that $\ell^\star \leq \min_{x \in S_n} |x|$. Hence any string of length strictly greater than $m_n \coloneqq \min_{x \in S_n} |x|$ belongs to *every* language consistent with $S_n$, and therefore belongs to $K$. One valid element-based generator outputs the first element in the canonical enumeration of $\mathcal{X}$ that has length at least $m_n$ and does not appear in $S_n$. However, this is not a valid set-based generator because it outputs only a single element rather than an infinite set. A valid set-based generator outputs all elements of $\mathcal{X}$ of length at least $m_n$ that do not appear in $S_n$.

### 2.2. Language Generation with Feedback

We now present two extensions of language generation that incorporate feedback. We begin with the model of query feedback introduced by Charikar & Pabbaraju (2025), which extends the basic model by allowing the generator to issue membership queries to the unknown target language. Fix a target language $K$ and an enumeration $x_1, x_2, \dots$ of $K$. Let $H_n = (x_1, \dots, x_n)$ and $S_n = \mathrm{set}(H_n)$. A *query-feedback generating algorithm* is a generating algorithm

$\mathcal{G} = (\mathcal{G}_n)_{n \in \mathbb{N}}$ that, at each step $n$, in addition to observing $x_n$, may adaptively choose a query string $y_n \in \mathcal{X}$ based on the ordered history $H_n$ and the previous query answers $a_{1:n-1}$, and then receive a response

$$a_n = \mathbb{1}\{y_n \in K\} \ .$$

The output of $\mathcal{G}_n$ at time $n$ may depend on the full ordered interaction history $(H_n, a_{1:n})$, and is interpreted exactly as in the basic model: if $\mathcal{G}$ is element-based it outputs a single string, and if it is set-based it outputs a (possibly infinite) set of strings.

**Definition 2** (Language Generation in the Limit with Query Feedback). *Consider the setting in Definition 1. The algorithm $\mathcal{G}$ is said to generate from $K$ in the limit with query feedback if, for all enumerations of $K$, there is some $n^\star \in \mathbb{N}$ such that for all $n \geq n^\star$:*

1. *if $\mathcal{G}$ is element-based, then $\mathcal{G}_n(H_n, a_{1:n}) \in K \setminus S_n$;*
2. *if $\mathcal{G}$ is set-based, then $\mathcal{G}_n(H_n, a_{1:n}) \subseteq K \setminus S_n$.*

*As before, $\mathcal{L}$ is generable in the limit with query feedback if there is an algorithm $\mathcal{G}$ that generates from $K$ in the limit with query feedback for every $K \in \mathcal{L}$.*

This model is inspired by classical works in learning theory, including the active learning model of Angluin (1987) which led to the celebrated $L^*$ algorithm for learning regular languages; for a survey of active learning settings, we refer the interested reader to Hanneke (2009).

**Mistake-feedback generation.** We now define the model of language generation in the limit with mistake feedback, inspired by Littlestone's classical online learning model (Littlestone, 1988). Here, the generator is not allowed to issue queries, but instead receives binary feedback indicating whether its most recent output was valid. This resembles how modern LLMs receive feedback: users have the option to indicate that they like or dislike the response of the model, thus sending it a binary signal. Fix a target language $K$ and an enumeration $x_1, x_2, \ldots$ of $K$, and let $H_n = (x_1, \ldots, x_n)$ and $S_n = \mathrm{set}(H_n)$. A *mistake-feedback generating algorithm* is a generating algorithm $\mathcal{G} = (\mathcal{G}_n)_{n \in \mathbb{N}}$ that, after producing its output at time $n$, receives a response $b_n$ defined as follows:

1. If $\mathcal{G}$ is element-based and outputs a string $z_n := \mathcal{G}_n(H_n, b_{1:n-1})$, then $b_n = \mathbb{1}\{z_n \in K \setminus S_n\}$;
2. If $\mathcal{G}$ is set-based and outputs a set $Z_n := \mathcal{G}_n(H_n, b_{1:n-1})$, then $b_n = \mathbb{1}\{z_n \in K \setminus S_n\}$ where $z_n$ is the first element of $Z_n$ according to the canonical enumeration over $\mathcal{X}$.

The algorithm at time $n + 1$ may depend on the full ordered interaction history $H_{n+1}$ and $(b_1, \ldots, b_n)$.

**Definition 3** (Language Generation in the Limit with Mistake Feedback). *Consider the setting in Definition 1. The algorithm $\mathcal{G}$ is said to generate from $K$ in the limit with mistake feedback if, for all enumerations of $K$, there exists $n^\star \in \mathbb{N}$ such that for all $n \geq n^\star$:*

1. *if $\mathcal{G}$ is element-based, then $\mathcal{G}_n(H_n, b_{1:n-1}) \in K \setminus S_n$;*
2. *if $\mathcal{G}$ is set-based, then $\mathcal{G}_n(H_n, b_{1:n-1}) \subseteq K \setminus S_n$.*

*As before, $\mathcal{L}$ is generable in the limit with mistake feedback if there is an algorithm $\mathcal{G}$ that generates from $K$ in the limit with mistake feedback for every $K \in \mathcal{L}$.*

A key distinction between the two feedback models is the nature of information received: In the query model, the generator *actively* chooses which strings to query before committing to an output, allowing it to "test" candidate strings for membership in $K$. In the mistake model, the generator receives feedback *passively* and only after committing to an output; moreover, this feedback reveals only whether the output was correct, not whether an arbitrary string belongs to $K$. Hence, a priori, the two models could have very different characterizations of generability—a question we revisit in Section 3.

## 3. Our Results

In this section, we present results and illustrate their broader applicability through several immediate implications.

**Outline of this Section.** We begin by introducing countable inner-covers (Definition 4), the key combinatorial notion underlying our characterizations (Section 3.1). We then present our characterization for mistake feedback (Theorem 3.1) along with the equivalence of element-based and set-based generators in this model (Theorem 3.2) in Section 3.2. Next, Section 3.3 covers query feedback, including the separation between element-based and set-based generators (Theorem 3.3) and the characterization for set-based generation (Theorem 3.4). Finally, Section 3.4 presents implications of our results, including closure properties (Section 3.4.1), generation with zero examples (Section 3.4.2), robustness to noisy feedback (Section 3.4.3), and consequences for generation without feedback (Section 3.4.4).

### 3.1. Countable Inner-Covers

A unifying theme across our characterizations is a new combinatorial notion that we call a *countable inner-cover*. Informally, a countable inner-cover of a collection $\mathcal{L}$ is a countable family of infinite sets such that every language in $\mathcal{L}$ contains at least one set from the family.

**Definition 4** (Countable Inner-Cover). *A collection $\mathcal{L}$ has a* countable inner-cover *if there exists a countable family $\mathcal{C} = \{C_1, C_2, \ldots\}$ of subsets of $\mathcal{X}$ such that:*

1. *$|C_i| = \infty$ for every $i \in \mathbb{N}$, and*

2. *for every $L \in \mathcal{L}$ there exists $i \in \mathbb{N}$ with $C_i \subseteq L$.*

In this case, we call $\mathcal{C}$ a countable inner-cover of $\mathcal{L}$. We emphasize that the requirement that each $C_i$ be infinite is essential: without it, every collection would trivially admit a countable inner-cover by taking $\mathcal{C} = \{\{x\} : x \in \mathcal{X}\}$, the collection of all singleton sets.

To gain some intuition, we consider several examples. First, note that all countable collections $\mathcal{L}$ have a countable inner-cover (we can simply set $\mathcal{C} = \mathcal{L}$). More interestingly, many uncountable collections also admit countable inner-covers:

- Consider a collection $\mathcal{L}_{\mathrm{AP}}$ over domain $\mathcal{X} = \mathbb{N}$ which consists of all languages that contain an infinite arithmetic progression. There are uncountably many such languages. However, the set of arithmetic progressions, which are countably many, form an inner cover of $\mathcal{L}_{\mathrm{AP}}$.
- As another example, let $\mathcal{L}_{\mathrm{Int}}$ over domain $\mathcal{X} = \mathbb{Q}$ consist of all languages that contain an open interval. While there are uncountably many such languages, the closed intervals over rationals form a countable inner-cover of $\mathcal{L}_{\mathrm{Int}}$.
- As a final example, any collection that can be generated from at most $N < \infty$ examples independent of the adversary—also called a *uniformly generable* collection—admits a countable inner-cover (see Section A.7). Roughly, this is because the generator can only see $|\mathcal{X}|^N$ possible inputs (which is countable) and then must generate; we can use this property to construct a countable inner cover.

On the other hand, natural families that do not have countable inner-covers also exist; for example, the collection of all infinite subsets of $\mathbb{N}$ has no such cover. A concrete counterexample appears in the proof of Theorem 3.3.

### 3.2. Characterization with Mistake Feedback

We now present our characterization for the mistake-feedback model. Its proof appears in Section A.1.

**Theorem 3.1** (Characterization of Mistake-feedback generation). *A collection $\mathcal{L}$ is generable in the limit with mistake feedback by an element-based generator (Definition 3) if and only if $\mathcal{L}$ has a countable inner-cover (Definition 4).*

Moreover, in the mistake-feedback model, set-based and element-based generators turn out to have exactly the same power. The proof of this result appears in Section A.1.

**Theorem 3.2** (Equivalence of set-based and element-based generators with mistake feedback). *A collection $\mathcal{L}$ is generable in the limit with mistake feedback by a set-based generator if and only if $\mathcal{L}$ is generable in the limit with mistake feedback by an element-based generator.*

Combining Theorems 3.1 and 3.2, we obtain a complete characterization for *both* notions of mistake-feedback gener-

ation. This equivalence contrasts with several other settings in language generation in the limit where set-based and element-based notions exhibit separations. For instance, Kleinberg & Wei (2025; 2026) showed that while element-based generators can always achieve density, set-based generators cannot. Similar differences were also shown in the work of (Mehrotra et al., 2025) in the context of noise tolerance. Moreover, this equivalence is not obvious from the definitions alone, since set-based generators must produce infinitely many valid strings each round rather than just one.

### 3.3. Results for Query Feedback

We now turn to query feedback (Definition 2), where the generator can adaptively query membership in the unknown target language. One might expect, by analogy with the mistake-feedback model, that set-based and element-based query-feedback generation are equivalent. Our first result shows that this is false. Its proof appears in Section A.2.

**Theorem 3.3** (Element-based vs. set-based non-equivalence under query feedback). *There exists a collection $\mathcal{L}$ such that $\mathcal{L}$ is generable in the limit with query feedback by an element-based generator, but $\mathcal{L}$ is not generable in the limit with query feedback by any set-based generator.*

At a high level, Theorem 3.3 exploits the fact that an element-based generator can use a membership query to "probe" a small amount of target structure and then output a single certified-correct string, without ever committing to an infinite set of such strings. In contrast, a set-based generator must eventually output an entire infinite subset of the target language at every round, which is a significantly more stringent requirement.

Despite this separation, set-based query-feedback generation admits a clean characterization. Strikingly, the characterizing condition is the same as for mistake-feedback generation. The proof of this result appears in Section A.2.

**Theorem 3.4** (Characterization of query-feedback set-generation). *A collection $\mathcal{L}$ is generable in the limit with query feedback by a set-based generator (Definition 2) if and only if $\mathcal{L}$ has a countable inner-cover (Definition 4).*

In particular, Theorem 3.4 together with Theorems 3.1 and 3.2 imply that *set-based* query-feedback generation is equivalent in power to mistake-feedback generation (either notion). This unified characterization reveals a surprising structure: despite the apparent differences between actively querying membership and passively receiving correctness feedback, the two models have identical expressive power for set-based generation.

**Remark 3.5** (Connections to LLM Training). In typical training pipelines, the model cannot adaptively query an oracle that answers whether an arbitrary string belongs to the target language (or distribution). By contrast, the

mistake-feedback model provides a simple binary signal indicating whether the model made an error on the current round, and this more closely mirrors the kind of evaluative feedback used in practice (for example, human preference labels, reward-model judgments, or automated validators all give feedback on the LLM's generated outputs rather than responses to queries issued by the LLM); even though, admittedly abstraction remains highly stylized. We therefore interpret Theorem 3.4 as evidence that, at least for set-based generation in the limit, access to membership queries does not yield strictly more power than access to mistake-style evaluative feedback.

## 3.4. Implications of Our Characterizations and Tools

One benefit of a characterization is that it can be used to derive nontrivial results more easily than proving them directly. A classical analogy is VC dimension: once we know that VC dimension characterizes PAC learnability, many closure properties such as closure under unions become immediate. We now highlight several such consequences of our results.

### 3.4.1. CLOSURE PROPERTIES

A striking difference between generation with and without feedback is closure under unions. In the basic model (without feedback), generation in the limit is *not* closed under finite unions: Hanneke et al. (2025); Bai et al. (2026) construct collections $\mathcal{L}_1, \mathcal{L}_2$ that are each generable, but whose union $\mathcal{L}_1 \cup \mathcal{L}_2$ is not. In contrast, our characterizations immediately imply union-closure for the feedback models studied here. Its proof appears in Section A.3.

**Corollary 3.6** (Closure under unions for feedback-based generation). *Let $\mathcal{L}_1, \mathcal{L}_2, \ldots$ be language collections. If each $\mathcal{L}_j$ is generable with mistake feedback (respectively, set-generable with query feedback), then $\bigcup_{j \in \mathbb{N}} \mathcal{L}_j$ is generable in the corresponding model.*

To see how this follows from the characterization, consider the mistake-feedback model. If each of $\mathcal{L}_1, \mathcal{L}_2, \ldots$ is generable, then by Theorem 3.1 each $\mathcal{L}_j$ has a countable inner-cover $\mathcal{C}_j$. From this, it follows that $\bigcup_{j \in \mathbb{N}} \mathcal{L}_j$ also admits the countable inner-cover, namely, $\bigcup_{j \in \mathbb{N}} \mathcal{C}_j$, and applying Theorem 3.1 again yields generability of the union. An analogous argument applies to the query-feedback model via Theorem 3.4; we formalize this in Corollary A.3.

In particular, this immediately recovers Bai et al. (2026)'s result that set-generation with membership queries is closed under finite unions, with a simpler proof.

More broadly, once one knows that countable inner-covers characterize these feedback models, many additional closure properties also become immediate: For example, if $\mathcal{L}_1$ and $\mathcal{L}_2$ admit countable inner-covers $\mathcal{C}_1$ and $\mathcal{C}_2$, then so does

their Cartesian product

$$\mathcal{L}_1 \otimes \mathcal{L}_2 := \left\{ \{(x_1, x_2) : x_1 \in L_1, \ x_2 \in L_2\} \right\}_{L_1 \in \mathcal{L}_1, \ L_2 \in \mathcal{L}_2},$$

with countable inner-cover $\mathcal{C}_1 \otimes \mathcal{C}_2$.[4] These corollaries illustrate the usefulness of our characterization.

### 3.4.2. GENERATION WITH ZERO EXAMPLES

Our characterizations show that in both feedback settings, the generator's *positive examples* from the adversary can be dispensed with entirely. Indeed, once a countable inner-cover exists, one can design generators that rely only on the feedback channel and do not use adversarial examples as an information source (aside from avoiding already-seen examples, as mandated by the model).

**Corollary 3.7** (Generation with zero-examples). *If $\mathcal{L}$ is generable with mistake feedback (respectively, set-generable with query feedback), then $\mathcal{L}$ is also generable with mistake feedback (respectively, set-generable with query feedback) with zero-examples (i.e., when $S_n = \emptyset$ for all $n$).*

In other words, the generator does not need access to the examples provided by the adversary; the feedback signal alone suffices. We proved this result by showing that if a collection has a countable inner-cover, then it can be generated with feedback without any examples; in fact this is already implicit in the proofs of our characterizations Theorems 3.1 and 3.4. The proof of Corollary 3.7 appears in Section A.4.

### 3.4.3. NOISY GENERATION WITH FEEDBACK

In several recent works, contamination and noise models for generation have been studied by allowing the *example stream* to contain errors of various kinds (Raman & Raman, 2025; Bai et al., 2026; Mehrotra et al., 2025). As seen in the previous section (Section 3.4.2), in our feedback settings adversary's examples turn out to be unnecessary, so we can handle *arbitrary* amounts of contamination in the adversarial examples.

A natural follow-up question is about corruption in the feedback channel itself. Here, we examine the following model:

**Definition 5** (Eventually correct feedback). *We say that the feedback transcript is* eventually correct *if there exists $n_0 \in \mathbb{N}$ such that all feedback bits after time $n_0$ are correct for the corresponding interaction (formalized in Definitions 6 and 7).*

In other words, the adversary may corrupt the feedback bits for a finite (but arbitrary) prefix of rounds, but must

---

[4]Indeed, to see this, fix any languages $L_1 \in \mathcal{L}_1$ and $L_2 \in \mathcal{L}_2$. Since $\mathcal{C}_1$ and $\mathcal{C}_2$ are inner covers, there exist $C_1 \in \mathcal{C}_1$ and $C_2 \in \mathcal{C}_2$ such that $C_1 \subseteq L_1$ and $C_2 \subseteq L_2$. Then $C_1 \times C_2 \subseteq L_1 \times L_2$. Since the choice of $L_1$ and $L_2$ was arbitrary, we can conclude that $\mathcal{C}_1 \otimes \mathcal{C}_2$ is an inner cover of $\mathcal{L}_1 \otimes \mathcal{L}_2$.

return correct bits thereafter. Our next result shows that this finite contamination does not change the class of generable collections. It appears in Section A.5.

**Corollary 3.8** (Characterization under eventually correct feedback). *A collection $\mathcal{L}$ is generable with eventually correct mistake feedback by an element-based generator (respectively, set-generable with eventually correct query feedback by a set-based generator) if and only if $\mathcal{L}$ has a countable inner-cover.*

This result shows that the countable-inner-cover condition is robust to any finite number of adversarial bit-flips in the feedback transcript, even when the example stream is arbitrarily corrupted (or entirely uninformative).

This robustness stands in sharp contrast to the no-feedback model, where even a single corrupted example can render a collection non-generable (Bai et al., 2026).

**Proof sketch of Corollary 3.8.** We derive Corollary 3.8 by combining our characterization with the explicit generators used in its proof. Starting from a countable inner-cover, the generator searches over cover elements and uses feedback to identify an inner set contained in the target; once the feedback becomes correct, this search stabilizes to an element of the cover and the generator behaves exactly as in the noiseless model. To absorb an arbitrary finite prefix of corrupted feedback, we wrap this search with the finite-expansion procedure of Mehrotra et al. (2025), which replaces each candidate set $C$ by all of its finite perturbations, thereby tolerating finitely many early inconsistencies while preserving the needed invariants. The key point is that the argument stays within *countable* objects: finite expansion preserves countability, so we retain a countable inner-cover and can reapply the characterization.

### 3.4.4. IMPLICATIONS FOR GENERATION WITHOUT FEEDBACK

Finally, we discuss implications of the technical tools developed for our main characterizations (Theorems 3.1 and 3.4) for the basic model of generation in the limit without feedback (Definition 1).

**Equivalence of element-based and set-based generation.** Kleinberg & Mullainathan (2024)'s original model outputs a single string each round, motivating the notion of element-based generation. Several subsequent works formulated set-based generation (also called auto-regressive generation) as a more LLM-like variant, and separations between the two notions were known in certain extensions (*e.g.*, under density or breadth constraints). However, whether these notions are equivalent in the basic "generation in the limit" model remained unclear. Our next theorem resolves this question. Its proof appears in Section A.6.

**Theorem 3.9** (Equivalence of set-based and element-based

generators). *A collection $\mathcal{L}$ is generable in the limit by a set-based generator if and only if $\mathcal{L}$ is generable in the limit by an element-based generator.*

This result follows by the same argument used to prove the earlier equivalence under mistake feedback (Theorem 3.2).

One might hope that, as in the mistake-feedback setting, generation in the basic (no-feedback) model could also be characterized by the existence of a countable inner-cover. Indeed, if $\mathcal{L}$ admits a countable inner-cover $\mathcal{C} = \{C_1, C_2, \ldots\}$, then it is tempting to run the classical generation algorithm of Kleinberg & Mullainathan (2024) on the *countable* family $\mathcal{C}$. The obstacle is that the unknown target language $L^\star \in \mathcal{L}$ need not belong to the cover $\mathcal{C}$; we only know that $C_i \subseteq L^\star$ for some $i$. Since the algorithm of Kleinberg & Mullainathan (2024) is designed for the setting where the target itself lies in the hypothesized countable collection, this mismatch causes the approach to break down.

One might then try to bypass this issue by appealing to other known algorithms for generation in the limit. However, we show that no such approach can yield a characterization purely in terms of countable inner-covers, via the following result, whose proof appears in Section A.6:

**Theorem 3.10** (Necessary and Sufficient Conditions). *The following statements hold for collections $\mathcal{L}$ in the no-feedback model.*

1. *If $\mathcal{L}$ has an inner-cover of finite size, then $\mathcal{L}$ is generable in the limit.*
2. *If $\mathcal{L}$ is generable in the limit, then $\mathcal{L}$ has a countable inner-cover (Definition 4).*

*Further, the following hold:*

1. *There is a collection $\mathcal{L}$ with a countable inner-cover that is not generable in the limit.*
2. *There is also a collection $\mathcal{L}$ that is generable in the limit and has a countable inner-cover.*
3. *There is also a collection $\mathcal{L}$ that is generable in the limit and does not have a finite inner-cover.*

In particular, Theorem 3.10 shows that generability in the no-feedback model is "sandwiched" between finite and countable inner-covers: having a finite inner-cover is sufficient for generability, and having a countable inner-cover is necessary, but neither condition fully characterizes generability. Closing this gap remains an interesting open problem.

## 4. Conclusion

In this work, we introduce countable inner-covers as a unifying combinatorial condition for feedback-based generation. Our main results show that a collection is generable with mistake feedback (Theorem 3.1) and set-generable with

query feedback (Theorem 3.4) if and only if it admits a countable inner-cover. In the mistake model, element-based and set-based generators are equivalent (Theorem 3.2); under query feedback they are not (Theorem 3.3). These characterizations also have several implications: closure under different operations (Section 3.4.1), generation with zero-examples (Section 3.4.2), robustness to finite corruption in feedback (Section 3.4.3), and, interestingly, also new results for the no-feedback setting (Section 3.4.4).

Our results raise several questions: First, what characterizes element-based query generation? The separation in Theorem 3.3 rules out countable inner-covers, but the characterization is unknown. Second, can one tolerate vanishing (rather than finite) corruption in the feedback channel while preserving generability, as in (Mehrotra et al., 2025)? Third, recent work (*e.g.*, (Kalavasis et al., 2025; Charikar & Pabbaraju, 2025)) has established strong impossibility results for achieving breadth (where breadth requires the generator to "cover" a large fraction of $K$) without feedback. Is language generation in the limit with breadth possible with feedback?

## Impact Statement

This paper presents work whose goal is to advance the field of Machine Learning. There are many potential societal consequences of our work, none which we feel must be specifically highlighted here.

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

# A. Proofs

In this section, we present the proofs of all our main results.

## A.1. Proof of Theorems 3.1 and 3.2 (Characterization with Mistake-Feedback)

In this section, we prove Theorems 3.1 and 3.2, which we restate below.

**Theorem 3.1** (Characterization of Mistake-feedback generation). *A collection $\mathcal{L}$ is generable in the limit with mistake feedback by an element-based generator (Definition 3) if and only if $\mathcal{L}$ has a countable inner-cover (Definition 4).*

**Theorem 3.2** (Equivalence of set-based and element-based generators with mistake feedback). *A collection $\mathcal{L}$ is generable in the limit with mistake feedback by a set-based generator if and only if $\mathcal{L}$ is generable in the limit with mistake feedback by an element-based generator.*

For technical reasons, we first prove Theorem 3.2 and then Theorem 3.1.

### A.1.1. PROOF OF THEOREMS 3.1 AND 3.2

*Proof.* We prove the two statements together. The only substantive change from the set-input version is that finite inputs to the generator are now ordered histories. The set $S_n$ is used only in the correctness requirement and in the explicit cover-based generators.

**Lemma A.1** (Locking histories for mistake-feedback generators). *Fix a target language $K \in \mathcal{L}$ and an element-based mistake-feedback generator $\mathcal{G}$ that generates from $K$ in the limit with mistake feedback. Then there exist $N \in \mathbb{N}$, a finite history $\sigma = (u_1, \ldots, u_N) \in K^N$, and the mistake-feedback transcript $\beta_{1:N-1}$ induced by running $\mathcal{G}$ on $\sigma$, such that for every finite continuation $\rho = (v_1, \ldots, v_r) \in K^r$ and every $r \geq 0$, if the feedback bits along $\sigma \parallel \rho$ are induced by the same mistake-feedback interaction, then*

$$\mathcal{G}_{N+r}(\sigma \parallel \rho, \beta_{1:N+r-1}) \in K \setminus \mathrm{set}(\sigma \parallel \rho).$$

*In words, after the finite ordered history $\sigma$, no continuation consisting of elements of $K$ can force another mistake.*

*Proof.* Suppose no such locking history exists. Then every finite history over $K$ that can arise in a run of $\mathcal{G}$ admits a finite continuation over $K$ after which $\mathcal{G}$ makes a mistake. Fix any enumeration $x_1, x_2, \ldots$ of $K$. We inductively build a single enumeration $y_1, y_2, \ldots$ of $K$ on which $\mathcal{G}$ makes infinitely many mistakes.

Let $\sigma_0$ be the empty history. At stage $i \geq 1$, append $x_i$ to the current history $\sigma_{i-1}$, obtaining $\sigma'_{i-1} = \sigma_{i-1} \parallel (x_i)$. Since $\sigma'_{i-1}$ is not locking, there is a finite continuation $\tau_i \in K^{r_i}$ such that, along the run on $\sigma'_{i-1} \parallel \tau_i$ with its induced feedback transcript, the generator makes a mistake at the final time of this extended history. Set

$$\sigma_i := \sigma'_{i-1} \parallel \tau_i.$$

The histories $\sigma_0, \sigma_1, \sigma_2, \ldots$ are nested. Hence they define an infinite sequence $y_1, y_2, \ldots$ whose prefix of length $|\sigma_i|$ is $\sigma_i$ for every $i$. Each $x_i$ appears in $y$ by the end of stage $i$, so $y$ is an enumeration of $K$.

Moreover, the run of $\mathcal{G}$ on $y$ agrees with the finite run on $\sigma_i$ up to time $|\sigma_i|$, including the induced feedback bits. Therefore $\mathcal{G}$ makes a mistake at time $|\sigma_i|$ for every $i$. This gives infinitely many mistakes on the enumeration $y$, contradicting that $\mathcal{G}$ generates from $K$ in the limit with mistake feedback. Thus a locking history exists. $\square$

**Element-based mistake feedback implies a countable inner-cover.** Let $\mathcal{G}$ be an element-based mistake-feedback generator for $\mathcal{L}$. For every finite history $\sigma \in \mathcal{X}^N$ and every bit string $\beta \in \{0,1\}^{N-1}$, define an infinite self-simulation as follows. Initialize $\sigma^{(0)} = \sigma$ and $\beta^{(0)} = \beta$. For $t \geq 1$, set

$$s_t(\sigma, \beta) := \mathcal{G}_{N+t-1}\Big(\sigma^{(t-1)}, \beta^{(t-1)}\Big)$$

and then update

$$\sigma^{(t)} := \sigma^{(t-1)} \parallel s_t(\sigma, \beta), \qquad \beta^{(t)} := \beta^{(t-1)} \parallel 1.$$

Let

$$C_{\sigma,\beta} := \{s_t(\sigma,\beta) : t \in \mathbb{N}\}.$$

There are only countably many pairs $(\sigma,\beta)$ because $\mathcal{X}$ is countable and $\sigma,\beta$ are finite. Hence the family of all infinite sets among the $C_{\sigma,\beta}$ is countable.

Fix any target $K \in \mathcal{L}$. By Lemma A.1, there is a locking history $\sigma \in K^N$ with induced transcript $\beta \in \{0,1\}^{N-1}$. Applying the locking property inductively to the continuations generated in the self-simulation gives

$$s_t(\sigma,\beta) \in K \setminus \mathrm{set}\left(\sigma^{(t-1)}\right) \qquad \text{for every } t \geq 1.$$

Thus the strings $s_1, s_2, \ldots$ are pairwise distinct and all lie in $K$, so $C_{\sigma,\beta}$ is an infinite subset of $K$. Therefore the countable family of infinite sets $C_{\sigma,\beta}$ is a countable inner-cover of $\mathcal{L}$.

**Countable inner-cover implies set-based mistake-feedback generation.** Conversely, suppose $\mathcal{L}$ has a countable inner-cover $\mathcal{C} = \{C_1, C_2, \ldots\}$. Let $\prec$ denote the canonical order on $\mathcal{X}$. Define a set-based mistake-feedback generator $\widehat{\mathcal{G}}$ as follows. On round $n$, after observing the ordered history $h_n = (x_1, \ldots, x_n)$ and feedback bits $b_{1:n-1}$, set $S_n = \mathrm{set}(h_n)$ and

$$i_n := 1 + \sum_{t=1}^{n-1} \mathbb{1}\{b_t = 0\}.$$

The generator outputs

$$\widehat{\mathcal{G}}_n(h_n, b_{1:n-1}) := C_{i_n} \setminus S_n.$$

This is an infinite set because $C_{i_n}$ is infinite and $S_n$ is finite. The feedback bit for this set is computed from the $\prec$-first element of $C_{i_n} \setminus S_n$.

Fix $K \in \mathcal{L}$ and an enumeration of $K$. Choose the least $i^\star$ such that $C_{i^\star} \subseteq K$. For each $i < i^\star$ with $C_i \not\subseteq K$, let

$$y_i := \min_\prec \{x \in C_i : x \notin K\} \qquad \text{and} \qquad P_i := \{x \in C_i : x \prec y_i\}.$$

The set $P_i$ is finite and is contained in $K$. Hence, after a finite time, every element of $P_i$ has appeared in $S_n$, while $y_i \notin S_n$. If the generator is still using index $i$ after that time, the first element of $C_i \setminus S_n$ is $y_i \notin K$, so the feedback bit is $0$ and the generator moves to the next index. Thus every index $i < i^\star$ is abandoned after finitely many rounds. Once $i_n = i^\star$, the first element of $C_{i^\star} \setminus S_n$ lies in $K \setminus S_n$, so the feedback bit is $1$ and the index never changes again. Consequently, for all sufficiently large $n$,

$$\widehat{\mathcal{G}}_n(h_n, b_{1:n-1}) = C_{i^\star} \setminus S_n \subseteq K \setminus S_n.$$

Hence $\mathcal{L}$ is set-generable with mistake feedback.

The same construction with singleton output

$$z_n := \min_\prec \{x \in C_{i_n} : x \notin S_n\}$$

gives the element-based generator used in Theorem 3.1.

**Set-based mistake feedback implies element-based mistake feedback.** Finally, let $\mathcal{G}$ be a set-based mistake-feedback generator. Define an element-based generator $\mathcal{G}'$ by outputting the first element of $\mathcal{G}$'s set in the canonical order:

$$\mathcal{G}'_n(h_n, b_{1:n-1}) := \mathsf{first}(\mathcal{G}_n(h_n, b_{1:n-1})).$$

The feedback bit received by $\mathcal{G}'$ is exactly the bit that $\mathcal{G}$ would have received, because the mistake-feedback rule for set-based generators also evaluates this first element. Therefore, once $\mathcal{G}_n(h_n, b_{1:n-1}) \subseteq K \setminus S_n$, its first element is also in $K \setminus S_n$. Thus set-based generability implies element-based generability.

Combining the three paragraphs proves both Theorem 3.1 and Theorem 3.2 in the sequence-input model. □

## A.2. Proof of Theorems 3.3 and 3.4 (Results for Query-Feedback)

In this section we prove Theorems 3.3 and 3.4, which we restate below.

**Theorem 3.3** (Element-based vs. set-based non-equivalence under query feedback). *There exists a collection $\mathcal{L}$ such that $\mathcal{L}$ is generable in the limit with query feedback by an element-based generator, but $\mathcal{L}$ is not generable in the limit with query feedback by any set-based generator.*

**Theorem 3.4** (Characterization of query-feedback set-generation). *A collection $\mathcal{L}$ is generable in the limit with query feedback by a set-based generator (Definition 2) if and only if $\mathcal{L}$ has a countable inner-cover (Definition 4).*

For technical reasons, we first present the proof of Theorem 3.4 and then the proof of Theorem 3.3.

### A.2.1. PROOF OF THEOREM 3.4

*Proof of Theorem 3.4.* We divide the proof into two parts corresponding to two directions of the claim.

**Part 1 (Set-based query-feedback generation implies a countable inner-cover):** Assume there exists a set-based query-feedback generator $\mathcal{G} = (\mathcal{G}_n)_{n \in \mathbb{N}}$ that generates from every $K \in \mathcal{L}$ in the sense of Definition 2. For each $n$, the output of $\mathcal{G}_n$ is determined by an input of the form $(h_n, a_{1:n})$, where $h_n \in \mathcal{X}^n$ is an ordered history and $a_{1:n} \in \{0,1\}^n$.

Since $\mathcal{X}$ is countable, the set $\mathcal{X}^n$ of length-$n$ histories is countable, and $\{0,1\}^n$ is finite. Hence the set of all possible inputs $(h_n, a_{1:n})$ is countable, and therefore the range

$$\mathsf{Range}(\mathcal{G}_n) := \{\mathcal{G}_n(h, a_{1:n}) : h \in \mathcal{X}^n, \ a_{1:n} \in \{0,1\}^n\}$$

is countable as well. Let

$$\mathcal{C} := \bigcup_{n \in \mathbb{N}} \mathsf{Range}(\mathcal{G}_n) \ .$$

Then $\mathcal{C}$ is a countable union of countable sets, hence countable. Enumerate its distinct elements as $\mathcal{C} = \{C_1, C_2, \ldots\}$. Since $\mathcal{G}$ is set-based, every $C_i$ is infinite.

We claim that $\mathcal{C}$ is an inner cover of $\mathcal{L}$. Fix any $K \in \mathcal{L}$ and any enumeration $x_1, x_2, \ldots$ of $K$, with $h_n := (x_1, \ldots, x_n)$ and $S_n := \mathsf{set}(h_n)$. Let $Z_n := \mathcal{G}_n(h_n, a_{1:n})$ denote the set output at time $n$ along this interaction. By Definition 2, there exists $n^\star$ such that for all $n \geq n^\star$, $Z_n \subseteq K \setminus S_n$. In particular, $Z_{n^\star} \subseteq K$. Since $Z_{n^\star} \in \mathcal{C}$, we have $Z_{n^\star} = C_i$ for some $i$, so $C_i \subseteq K$. As $K$ was arbitrary, $\mathcal{C}$ is a countable inner-cover.

**Part 2 (Countable inner-cover implies set-based query-feedback generation).** Assume $\mathcal{L}$ has a countable inner-cover $\mathcal{C} = \{C_1, C_2, \ldots\}$. Let $\prec$ be the canonical order on $\mathcal{X}$ induced by $(\bar{x}_1, \bar{x}_2, \ldots)$. We construct a set-based query-feedback generator $\mathcal{G}$. At time $n$, given the ordered history $h_n$ and past answers $a_{1:n-1}$, set $S_n := \mathsf{set}(h_n)$ and define the current index $i_n$ to be one more than the number of negative query answers so far, *i.e.*,

$$i_n := 1 + \sum_{t=1}^{n-1} \mathbb{1}\{a_t = 0\} \ .$$

Choose the query point $y_n$ as the smallest element of $C_{i_n}$ that has not been seen in the input history, *i.e.*,

$$y_n := \min_{\prec}\{x \in C_{i_n} : x \notin S_n\} \ ,$$

which is well-defined because $S_n$ is finite and $C_{i_n}$ is infinite. After receiving the membership answer $a_n = \mathbb{1}\{y_n \in K\}$, output the set

$$Z_n := \begin{cases} C_{i_n} \setminus S_n & \text{if } a_n = 1 \,, \\ C_{i_n+1} \setminus S_n & \text{if } a_n = 0 \,. \end{cases}$$

Each $Z_n$ is infinite, since it is an infinite set minus a finite set, so $\mathcal{G}$ is set-based.

Fix an arbitrary target $K \in \mathcal{L}$ and an arbitrary enumeration $x_1, x_2, \ldots$ of $K$, with histories $h_n = (x_1, \ldots, x_n)$ and seen sets $S_n = \mathsf{set}(h_n)$. Choose the least $i^\star$ such that $C_{i^\star} \subseteq K$; such an index exists since $\mathcal{C}$ is an inner cover. We show that there exists $n^\star$ such that for all $n \geq n^\star$, $Z_n \subseteq K \setminus S_n$.

First consider any $i < i^\star$. Since $C_i \not\subseteq K$, the set $C_i \setminus K$ is nonempty; let

$$u_i := \min_{\prec}\{x \in C_i : x \notin K\} \qquad \text{and} \qquad P_i := \{x \in C_i : x \prec u_i\}\,.$$

Because $\prec$ is induced by an enumeration of $\mathcal{X}$, the initial segment $\{x \in \mathcal{X} : x \prec u_i\}$ is finite, so $P_i$ is finite. By minimality of $u_i$, every element of $P_i$ lies in $K$, hence there exists $N_i$ such that $P_i \subseteq S_{N_i}$. For any $n \geq N_i$, if the current index satisfies $i_n = i$, then $P_i \subseteq S_n$ and $u_i \notin S_n$ (since $u_i \notin K$), so

$$y_n = \min_{\prec}\{x \in C_i : x \notin S_n\} = u_i\,.$$

Therefore $a_n = \mathbb{1}\{u_i \in K\} = 0$, and the update rule forces the index to increase after round $n$. In particular, $\mathcal{G}$ cannot remain at any index $i < i^\star$ forever. Since there are only finitely many indices less than $i^\star$, there exists a finite time $n^\star$ after which $i_n \geq i^\star$ for all $n \geq n^\star$.

Next, once $i_n = i^\star$ at some time, then for all later times the index never increases: indeed, when $i_n = i^\star$, we have $y_n \in C_{i^\star} \setminus S_n \subseteq K$, so $a_n = 1$, and the index does not change. Thus, for all sufficiently large $n$, we have $i_n = i^\star$ and $a_n = 1$, hence

$$Z_n = C_{i^\star} \setminus S_n \subseteq K \setminus S_n\,.$$

This is exactly the success condition in Definition 2 for set-based generators. Since $K$ and its enumeration were arbitrary, $\mathcal{G}$ generates from every $K \in \mathcal{L}$ in the limit with query feedback. $\qquad\square$

### A.2.2. PROOF OF THEOREM 3.3

*Proof of Theorem 3.3.* Let $\mathcal{X} = \mathbb{N}$ with the canonical order $1, 2, \ldots$ denoted by $\prec$. For each $j \in \mathbb{N}$, define the block

$$B_j := \{3j - 2,\ 3j - 1,\ 3j\}\,.$$

For each infinite bit-sequence $\sigma \in \{0, 1\}^{\mathbb{N}}$, define a language $L_\sigma \subseteq \mathbb{N}$ by

$$L_\sigma := \bigcup_{j \in \mathbb{N}} \begin{cases} \{3j - 2,\ 3j - 1\} & \text{if } \sigma_j = 1, \\ \{3j\} & \text{if } \sigma_j = 0. \end{cases}$$

In words, we partition $\mathbb{N}$ into consecutive blocks $B_j = \{3j - 2, 3j - 1, 3j\}$. The bit $\sigma_j$ determines which strings from the $j$th block belong to $L_\sigma$: if $\sigma_j = 1$ then $L_\sigma$ contains the first two elements $3j - 2$ and $3j - 1$ but excludes $3j$, while if $\sigma_j = 0$ then $L_\sigma$ contains only the last element $3j$ and excludes $3j - 2$ and $3j - 1$. Thus, each block contributes either a "pair" $\{3j - 2, 3j - 1\}$ or a "singleton" $\{3j\}$, and the sequence $\sigma$ specifies this choice independently for every block. Let

$$\mathcal{L} := \{L_\sigma : \sigma \in \{0, 1\}^{\mathbb{N}}\}\,.$$

Note that $\mathcal{L}$ has uncountably many languages.

We will show that $\mathcal{L}$ is element-generable with query feedback (Step 1) but has no countable inner-cover (Step 2) and so is not set-generable with query feedback (due to Theorem 3.4), which together imply the desired theorem.

**Step 1 ($\mathcal{L}$ is element-based generable with query feedback):** We describe an element-based query-feedback generator $\mathcal{G}$. At time $n$, given the ordered history $h_n$ with $S_n = \text{set}(h_n)$ and past answers $a_{1:n-1}$, let $t_{n-1}$ denote the largest block-index used previously by the algorithm (with $t_0 := 0$). Define

$$j_n := \min\{j > t_{n-1} : B_j \cap S_n = \emptyset\}\,.$$

Such a $j_n$ exists because $S_n$ is finite and the blocks $\{B_j\}_{j \in \mathbb{N}}$ partition $\mathbb{N}$. The algorithm issues the query

$$y_n := 3j_n - 2, \qquad \text{and receives} \qquad a_n = \mathbb{1}\{y_n \in K\}\,,$$

where $K$ is the unknown target language. It then outputs the element

$$z_n := \begin{cases} 3j_n - 1 & \text{if } a_n = 1, \\ 3j_n & \text{if } a_n = 0. \end{cases}$$

Intuitively, in each fresh block $B_{j_n} = \{3j_n - 2, 3j_n - 1, 3j_n\}$ the generator first queries the "indicator" element $3j_n - 2$ to learn which of the two possible patterns the target language uses on that block. If the query answer is positive ($a_n = 1$), then the block must be of the "pair" type and the generator safely outputs the other member of the pair, $3j_n - 1$. If the answer is negative ($a_n = 0$), then the block must be of the "singleton" type and the generator outputs the unique in-language element $3j_n$. In both cases, the output is guaranteed to lie in the target language and is new because the block was chosen to be unseen.

Formally, we can verify the correctness of the generator as follows: Fix any target $K = L_\sigma \in \mathcal{L}$ and any enumeration $x_1, x_2, \ldots$ of $K$ with $h_n = (x_1, \ldots, x_n)$ and $S_n = \mathrm{set}(h_n)$. By construction, $B_{j_n} \cap S_n = \emptyset$, hence $z_n \notin S_n$. Moreover, $a_n = 1$ if and only if $3j_n - 2 \in L_\sigma$, which holds if and only if $\sigma_{j_n} = 1$. If $\sigma_{j_n} = 1$ then $3j_n - 1 \in L_\sigma$, and if $\sigma_{j_n} = 0$ then $3j_n \in L_\sigma$. Therefore in all cases $z_n \in K$. We conclude that for every $n$,

$$\mathcal{G}_n(h_n, a_{1:n}) = z_n \in K \setminus S_n,$$

so $\mathcal{G}$ generates from $K$ in the limit with query feedback with $n^\star = 1$. Since $K$ was arbitrary, $\mathcal{L}$ is element-based generable in the limit with query feedback. (Additionally, the queries do not repeat since $j_n$ is strictly increasing.)

**Step 2 ($\mathcal{L}$ has no countable inner-cover):** We show that $\mathcal{L}$ does not admit a countable inner-cover (Definition 4). Assume toward a contradiction that $\mathcal{C} = \{C_1, C_2, \ldots\}$ is a countable inner-cover of $\mathcal{L}$. Fix $i \in \mathbb{N}$. Since $C_i$ is infinite and each block $B_j$ is finite, $C_i$ intersects infinitely many blocks: in other words, the following set is infinite

$$J_i := \{j \in \mathbb{N} : C_i \cap B_j \neq \emptyset\} \, .$$

Construct inductively a sequence of distinct indices $j_1, j_2, \ldots$ such that $j_i \in J_i$ for each $i$: given $j_1, \ldots, j_{i-1}$, choose $j_i \in J_i \setminus \{j_1, \ldots, j_{i-1}\}$, which is possible since $J_i$ is infinite and $\{j_1, \ldots, j_{i-1}\}$ is finite. For each $i$, pick an element

$$e_i \in C_i \cap B_{j_i} \, .$$

Now define a bit-sequence $\sigma \in \{0, 1\}^{\mathbb{N}}$ by specifying the bits at the selected coordinates: for each $i$ set

$$\sigma_{j_i} := \begin{cases} 1 & \text{if } e_i = 3j_i, \\ 0 & \text{if } e_i \in \{3j_i - 2, 3j_i - 1\}, \end{cases}$$

and set $\sigma_j := 0$ for all other $j$.

By construction, $e_i \in C_i$. We claim that $e_i \notin L_\sigma$ for every $i$. Indeed, if $e_i = 3j_i$ then $\sigma_{j_i} = 1$ and the block-contribution to $L_\sigma$ at $j_i$ is $\{3j_i - 2, 3j_i - 1\}$, excluding $3j_i$; if $e_i \in \{3j_i - 2, 3j_i - 1\}$ then $\sigma_{j_i} = 0$ and the block-contribution is $\{3j_i\}$, excluding $3j_i - 2$ and $3j_i - 1$. Thus, $e_i \notin L_\sigma$ in all cases. Consequently, there is no $i$ such that

$$C_i \subseteq L_\sigma \, ,$$

since $e_i \in C_i$ but $e_i \notin L_\sigma$. This contradicts that $\mathcal{C}$ is an inner cover (it fails on the language $L_\sigma \in \mathcal{L}$). Therefore, $\mathcal{L}$ has no countable inner-cover.

$\square$

### A.3. Proof of Corollary 3.6 (Closure Properties under Unions)

In this section we prove Corollary 3.6, which we restate below.

**Corollary 3.6** (Closure under unions for feedback-based generation)**.** *Let $\mathcal{L}_1, \mathcal{L}_2, \ldots$ be language collections. If each $\mathcal{L}_j$ is generable with mistake feedback (respectively, set-generable with query feedback), then $\bigcup_{j \in \mathbb{N}} \mathcal{L}_j$ is generable in the corresponding model.*

We divide the proof into the following two corollaries corresponding to the two models of feedback.

**Corollary A.2** (Closure under countable unions under mistake feedback)**.** *Let $\mathcal{L}_1, \mathcal{L}_2, \ldots \subseteq 2^{\mathcal{X}}$ be language collections. If each $\mathcal{L}_j$ is generable in the limit with mistake feedback by an element-based generator, then $\bigcup_{j \in \mathbb{N}} \mathcal{L}_j$ is generable in the limit with mistake feedback by an element-based generator.*

*Proof.* For each $j \in \mathbb{N}$, Theorem 3.1 implies that $\mathcal{L}_j$ admits a countable inner-cover $\mathcal{C}^{(j)} = \{C_1^{(j)}, C_2^{(j)}, \ldots\}$ (Definition 4). Let $\mathcal{C} := \bigcup_{j \in \mathbb{N}} \mathcal{C}^{(j)}$. Then $\mathcal{C}$ is countable because it is a countable union of countable sets. Further, every member of $\mathcal{C}$ is infinite. If $L \in \bigcup_{j \in \mathbb{N}} \mathcal{L}_j$, then $L \in \mathcal{L}_{j^\star}$ for some $j^\star$, so there exists $C \in \mathcal{C}^{(j^\star)} \subseteq \mathcal{C}$ with $C \subseteq L$. Hence $\mathcal{C}$ is a countable inner cover of $\bigcup_{j \in \mathbb{N}} \mathcal{L}_j$. Now Theorem 3.1 implies $\bigcup_{j \in \mathbb{N}} \mathcal{L}_j$ is generable in the limit with mistake feedback by an element-based generator. $\qquad\square$

**Corollary A.3** (Closure under countable unions under query feedback). *Let $\mathcal{L}_1, \mathcal{L}_2, \ldots \subseteq 2^{\mathcal{X}}$ be language collections. If each $\mathcal{L}_j$ is generable in the limit with query feedback by a set-based generator, then $\bigcup_{j \in \mathbb{N}} \mathcal{L}_j$ is generable in the limit with query feedback by a set-based generator.*

*Proof.* The proof is identical to the proof of Corollary A.2 except using Theorem 3.4 instead of Theorem 3.1. $\qquad\square$

## A.4. Proof of Corollary 3.7 (Generation with Zero Examples)

In this section, we prove Corollary 3.7, which we restate below.

**Corollary 3.7** (Generation with zero-examples). *If $\mathcal{L}$ is generable with mistake feedback (respectively, set-generable with query feedback), then $\mathcal{L}$ is also generable with mistake feedback (respectively, set-generable with query feedback) with zero-examples (i.e., when $S_n = \emptyset$ for all $n$).*

We divide Corollary 3.7 into the following two results corresponding to the feedback model.

**Corollary A.4** (Mistake feedback does not require adversarial examples). *If a collection $\mathcal{L}$ is generable in the limit with mistake feedback (Definition 3), then $\mathcal{L}$ is also generable in the limit with mistake feedback in the* zero-example *setting where the adversary provides no examples, that is, along the interaction in which $S_n = \emptyset$ for all $n$ (equivalently, the generator receives only the feedback bits and no positive samples).*

If one wishes to conclude the same statement for set-based generators, it follows from Theorem 3.2.

**Corollary A.5** (Query feedback does not require adversarial examples). *If a collection $\mathcal{L}$ is set-generable in the limit with query feedback (Definition 2), then $\mathcal{L}$ is also set-generable in the limit with query feedback in the* zero-example *setting where the adversary provides no examples, that is, along the interaction in which $S_n = \emptyset$ for all $n$ (equivalently, the generator receives only query answers and no positive samples).*

In the remainder of this section, we prove the above two results.

### A.4.1. PROOF OF COROLLARY A.4

*Proof of Corollary A.4.* By Theorem 3.1, the assumption that $\mathcal{L}$ is generable in the limit with mistake feedback implies that $\mathcal{L}$ has a countable inner-cover $\mathcal{C} = \{C_1, C_2, \ldots\}$ as in Definition 4. We now give an element-based mistake-feedback generator that uses *no* adversarial examples. Let $\prec$ be the canonical order on $\mathcal{X}$ induced by $(\overline{x}_1, \overline{x}_2, \ldots)$. On round $n$, the generator has access only to the past feedback bits $b_{1:n-1}$, and it can reconstruct its own past outputs; let

$$T_n := \{z_1, \ldots, z_{n-1}\}$$

denote the set of previously output strings (so $|T_n| \leq n - 1$). Define the current index $i_n$ to be one more than the number of mistakes made so far, *i.e.*,

$$i_n := 1 + \sum_{t=1}^{n-1} \mathbb{1}\{b_t = 0\},$$

and output $z_n$ to be the smallest element of $C_{i_n}$ that has not been seen in the input $S_n$, *i.e.*,

$$z_n := \min_{\prec}\{x \in C_{i_n} : x \notin T_n\}.$$

This is well-defined because $T_n$ is finite and $C_{i_n}$ is infinite. In the zero-example setting we have $S_n = \emptyset$, so the mistake-feedback bit equals

$$b_n = \mathbb{1}\{z_n \in K \setminus S_n\} = \mathbb{1}\{z_n \in K\}.$$

Fix an arbitrary target $K \in \mathcal{L}$. Choose the least $i^\star$ such that $C_{i^\star} \subseteq K$, which exists since $\mathcal{C}$ is an inner cover. By noting that the above element-based generator is the same as the element-based generator in Step 2 of the proof of Theorem 3.1, it follows that the above generator eventually reaches index $i^\star$ and thereafter outputs only elements of $K$. For completeness, we reprove this below.

Consider any $i < i^\star$. Since $C_i \nsubseteq K$, the set $C_i \setminus K$ is nonempty; let

$$u_i := \min_\prec \{x \in C_i : x \notin K\} \qquad \text{and} \qquad P_i := \{x \in C_i : x \prec u_i\}.$$

Because $\prec$ is induced by an enumeration of $\mathcal{X}$, the initial segment $\{x \in \mathcal{X} : x \prec u_i\}$ is finite, hence $P_i$ is finite. While the generator is at index $i$, it outputs the $\prec$-least element of $C_i$ that it has not output before. Therefore, after at most $|P_i| + 1$ rounds spent at index $i$, it outputs $u_i$. Since $u_i \notin K$, this yields feedback $b_n = 0$, so the index increases and the generator leaves $i$ after finitely many steps. Since there are only finitely many indices smaller than $i^\star$, there exists a finite round $n^\star$ such that for all $n \geq n^\star$ we have $i_n = i^\star$. For every $n \geq n^\star$, the output satisfies

$$z_n \in C_{i^\star} \subseteq K,$$

so $b_n = 1$ forever and the index never changes again. Thus, in the zero-example setting, there exists $n^\star$ such that for all $n \geq n^\star$, $z_n \in K \setminus S_n = K$. This is exactly generation in the limit with mistake feedback (with $S_n = \emptyset$ for all $n$).

$\square$

### A.4.2. PROOF OF COROLLARY A.5

*Proof of Corollary A.5.* By Theorem 3.4, the assumption that $\mathcal{L}$ is set-generable in the limit with query feedback implies that $\mathcal{L}$ has a countable inner-cover $\mathcal{C} = \{C_1, C_2, \ldots\}$ as in Definition 4. We now give a set-based query-feedback generator that uses *no* adversarial examples. The proof uses the construction from Part 2 of Theorem 3.4, and we include it here for completeness.

Let $\prec$ be the canonical order on $\mathcal{X}$ induced by $(\overline{x}_1, \overline{x}_2, \ldots)$. In the zero-example setting we have $S_n = \emptyset$ for all $n$. On round $n$, the generator has access to the past answers $a_{1:n-1}$ and can reconstruct its own past queries. Define the current index $i_n$ to be one more than the number of negative answers obtained so far, *i.e.*,

$$i_n := 1 + \sum_{t=1}^{n-1} \mathbb{1}\{a_t = 0\}.$$

Let

$$T_n := \{y_t : t \leq n - 1 \text{ and } i_t = i_n\}$$

be the set of query points previously asked *while at the current index $i_n$*. (As $i_n$ is nondecreasing, $T_n$ is finite and is determined by $a_{1:n-1}$.) Define the query point

$$y_n := \min_\prec \{x \in C_{i_n} : x \notin T_n\},$$

which is well-defined because $T_n$ is finite and $C_{i_n}$ is infinite. After receiving the membership answer $a_n = \mathbb{1}\{y_n \in K\}$, output the set

$$Z_n := \begin{cases} C_{i_n} \setminus T_n & \text{if } a_n = 1, \\ C_{i_n+1} & \text{if } a_n = 0. \end{cases}$$

Each $Z_n$ is infinite, hence $\mathcal{G}$ is set-based.

Fix an arbitrary target $K \in \mathcal{L}$. Choose the least $i^\star$ such that $C_{i^\star} \subseteq K$, which exists since $\mathcal{C}$ is an inner cover. We show that there exists $n^\star$ such that for all $n \geq n^\star$, $Z_n \subseteq K \setminus S_n = K$. Consider any $i < i^\star$. Since $C_i \nsubseteq K$, the set $C_i \setminus K$ is nonempty; let

$$u_i := \min_\prec \{x \in C_i : x \notin K\} \qquad \text{and} \qquad P_i := \{x \in C_i : x \prec u_i\}.$$

Because $\prec$ is induced by an enumeration of $\mathcal{X}$, the initial segment $\{x \in \mathcal{X} : x \prec u_i\}$ is finite, so $P_i$ is finite. While the generator is at index $i$, it queries the $\prec$-least element of $C_i$ that it has not queried before (that is, the $\prec$-least element of

$C_i \setminus T_n$). Therefore, after at most $|P_i| + 1$ rounds spent at index $i$, it queries $u_i$. Since $u_i \notin K$, this yields $a_n = 0$, so the index increases and the generator leaves $i$ after finitely many steps. Since there are only finitely many indices smaller than $i^\star$, there exists a finite round $n^\star$ such that for all $n \geq n^\star$ we have $i_n = i^\star$.

For every $n \geq n^\star$, we have $i_n = i^\star$. Moreover, since $C_{i^\star} \subseteq K$, every query point $y_n \in C_{i^\star}$ satisfies $a_n = 1$, so the index never changes again. Thus, for all $n \geq n^\star$, $Z_n = C_{i^\star} \setminus T_n \subseteq C_{i^\star} \subseteq K = K \setminus S_n$. This is exactly the success condition in Definition 2 for set-based generators in the zero-example setting. Since $K$ was arbitrary, $\mathcal{G}$ set-generates $\mathcal{L}$ in the limit with query feedback without using any adversarial examples. □

### A.5. Proof of Corollary 3.8 (Noisy Generation with Feedback)

In this section, we prove Corollary 3.8 which we restate below.

**Corollary 3.8** (Characterization under eventually correct feedback). *A collection $\mathcal{L}$ is generable with eventually correct mistake feedback by an element-based generator (respectively, set-generable with eventually correct query feedback by a set-based generator) if and only if $\mathcal{L}$ has a countable inner-cover.*

We divide Corollary 3.8 into the following two theorems.

**Theorem A.6** (Characterization under eventually correct mistake feedback). *A collection $\mathcal{L}$ is generable in the limit with eventually correct mistake feedback (Definition 6) by an element-based generator if and only if $\mathcal{L}$ has a countable inner cover (Definition 4).*

Theorem A.6 shows that finite contamination in the feedback transcript does not reduce the power of the mistake-feedback model: the class of collections that are generable in the limit is exactly the same as in the noiseless setting. Equivalently, the countable-inner-cover condition is robust to any finite number of adversarial corruptions of the feedback bits, even when combined with an arbitrarily corrupted example stream.

**Theorem A.7** (Characterization under eventually correct query feedback). *A collection $\mathcal{L}$ is set-generable in the limit with eventually correct query feedback (Definition 7) by a set-based generator if and only if $\mathcal{L}$ has a countable inner cover (Definition 4).*

For clarity, we formally instantiate the eventually correct feedback model (Definition 5) for both feedback models below.

**Definition 6** (Eventually correct mistake feedback). *Consider the mistake-feedback interaction model of Definition 3, except that the generator observes a (possibly contaminated) feedback transcript $(\widetilde{b}_1, \widetilde{b}_2, \ldots) \in \{0,1\}^{\mathbb{N}}$. Let $(b_1, b_2, \ldots) \in \{0,1\}^{\mathbb{N}}$ be the "correct" information bits. We say the feedback is* eventually correct *if there exists $n_0 \in \mathbb{N}$ such that for all $n \geq n_0$, the observed bit $\widetilde{b}_n$ equals the correct mistake-feedback bit prescribed by Definition 3 for that round.*

**Definition 7** (Eventually correct query feedback). *Consider the query-feedback interaction model of Definition 2, except that the generator observes a (possibly contaminated) query-answer transcript $(\widetilde{a}_1, \widetilde{a}_2, \ldots) \in \{0,1\}^{\mathbb{N}}$. Let $(a_1, a_2, \ldots) \in \{0,1\}^{\mathbb{N}}$ be the "correct" query-answer bits, i.e., the bits that would be returned under Definition 2 for that interaction. We say the feedback is* eventually correct *if there exists $n_0 \in \mathbb{N}$ such that for all $n \geq n_0$, the observed bit $\widetilde{a}_n$ equals the correct query-answer bit prescribed by Definition 2 for that round.*

In the remainder of this section, we prove Theorems A.6 and A.7.

*Proof of Theorem A.6.* We divide the proof into two parts.

**Part 1 (Countable covers implies $\mathcal{L}$ is generable):** Assume that $\mathcal{L}$ has a countable inner cover $\mathcal{C} = \{C_1, C_2, \ldots\}$. We prove that $\mathcal{L}$ is generable in the limit with eventually correct mistake feedback.

**Step 1.1: Finite expansion preserves countable inner covers.** We first use the finite expansion subroutine of Mehrotra et al. (2025) to expand the collection $\mathcal{L}$ into a collection $\widetilde{\mathcal{L}}$ as follows:

$$\widetilde{\mathcal{L}} := \{ L \cup A \setminus B : L \in \mathcal{L}, \; A \subseteq \mathcal{X}, \; B \subseteq \mathcal{X}, \; |A| < \infty, \; |B| < \infty \}.$$

In words, $\widetilde{\mathcal{L}}$ is the collection obtained from $\mathcal{L}$ by taking every $L \in \mathcal{L}$ and adding and/or removing finitely many elements (that is, including every finite symmetric-difference variant of $L$). Similarly, we define the corresponding finite expansion of the cover $\mathcal{C}$ by

$$\widetilde{\mathcal{C}} := \{\, C_i \cup A \setminus B : i \in \mathbb{N},\ A \subseteq \mathcal{X},\ B \subseteq \mathcal{X},\ |A| < \infty,\ |B| < \infty \,\}\,.$$

As with $\widetilde{\mathcal{L}}$, $\widetilde{\mathcal{C}}$ is the collection obtained from the cover $\mathcal{C}$ by taking every $C_i \in \mathcal{C}$ and adding and/or removing finitely many elements (that is, including every finite symmetric-difference variant of each $C_i$). Since $\mathcal{X}$ is countable, the collection of finite subsets of $\mathcal{X}$ is countable. Therefore the set of pairs $(A, B)$ of finite subsets is countable, and hence $\widetilde{\mathcal{C}}$ is countable as a countable union of countable sets. Moreover, every set in $\widetilde{\mathcal{C}}$ is infinite, since it is obtained from an infinite $C_i$ by adding and removing only finitely many elements.

We claim that $\widetilde{\mathcal{C}}$ is a countable inner cover of $\widetilde{\mathcal{L}}$. To see this, fix any $\widetilde{K} \in \widetilde{\mathcal{L}}$. By definition, $\widetilde{K} = K \cup A \setminus B$ for some $K \in \mathcal{L}$ and finite sets $A, B$. Choose $i^\star$ such that $C_{i^\star} \subseteq K$ (which exists since $\mathcal{C}$ is an inner cover). Then $C_{i^\star} \setminus B \subseteq K \setminus B \subseteq K \cup A \setminus B = \widetilde{K}$. Since $C_{i^\star} \setminus B \in \widetilde{\mathcal{C}}$, this proves the covering property.

**Step 1.2 (Reduce contaminated feedback to noiseless feedback for an expanded target).** Fix an arbitrary target $K \in \mathcal{L}$ and consider an arbitrary interaction in the sense of Definition 3, except that the generator receives an eventually correct feedback transcript $\widetilde{b}_1, \widetilde{b}_2, \dots$ as in Definition 5. We focus on generators that ignore the adversary's examples as a source of information and use only the feedback bits, while still avoiding elements in $S_n$ as required by the model.

Let $\prec$ denote the canonical order on $\mathcal{X}$. We define an element-based generator $\mathcal{G}$ that proceeds in *phases* indexed by $i \in \mathbb{N}$, and that never outputs the same string twice. Formally, on round $n$, let

$$T_n := \{z_1, \dots, z_{n-1}\}$$

be the set of previously output strings. Define the current phase index from the observed feedback bits by

$$i_n := 1 + \sum_{t=1}^{n-1} \mathbb{1}\{\widetilde{b}_t = 0\}\,.$$

Output the smallest element of $\widetilde{C}_{i_n}$ which has not been output so far (*i.e.*, not appeared in $T_n$) and has also not appeared in the seen set $S_n$ so far, *i.e.*,

$$z_n := \min_{\prec}\{x \in \widetilde{C}_{i_n} : x \notin S_n \cup T_n\}\,.$$

Here, $\widetilde{C}_i$ denotes the $i$-th set in some fixed enumeration of $\widetilde{\mathcal{C}}$. $z_n$ is well-defined because $S_n \cup T_n$ is finite and $\widetilde{C}_{i_n}$ is infinite. Since $z_n \notin S_n$ by construction, the *correct* mistake-feedback bit at time $n$ equals $b_n = \mathbb{1}\{z_n \in K\}$. By eventual correctness, there exists $n_0$ such that for all $n \geq n_0$,

$$\widetilde{b}_n = b_n = \mathbb{1}\{z_n \in K\}\,.$$

We now define a finite modification $\widetilde{K}$ of $K$ so that the *entire* transcript $\widetilde{b}_1, \widetilde{b}_2, \dots$ is consistent with correct mistake feedback for $\widetilde{K}$. Let

$$A := \{z_n : n < n_0,\ \widetilde{b}_n = 1,\ z_n \notin K\} \qquad \text{and} \qquad B := \{z_n : n < n_0,\ \widetilde{b}_n = 0,\ z_n \in K\}\,.$$

Both $A$ and $B$ are finite (they are subsets of $\{z_1, \dots, z_{n_0-1}\}$). Define

$$\widetilde{K} := (K \cup A) \setminus B.$$

Then $\widetilde{K} \in \widetilde{\mathcal{L}}$ by definition of finite expansion, and by construction we have that for each $n \in \mathbb{N}$

$$\widetilde{b}_n = \mathbb{1}\Big\{z_n \in \widetilde{K}\Big\}\,.$$

(For $n \geq n_0$ this holds because $\widetilde{K}$ agrees with $K$ outside the finite set $A \cup B$; for $n < n_0$ it holds by the definition of $A$ and $B$.) Thus, if one views the same interaction as having target language $\widetilde{K}$, then the observed transcript $\widetilde{b}$ is *precisely* the correct mistake-feedback transcript for $\widetilde{K}$. Equivalently, the generator $\mathcal{G}$ is experiencing a noiseless mistake-feedback interaction with some target $\widetilde{K} \in \widetilde{\mathcal{L}}$.

**Step 1.3 (Generating $\widetilde{K}$ in the noiseless model and concluding generation for $K$):** By Step 1.1, $\widetilde{\mathcal{L}}$ has a countable inner cover (namely $\widetilde{\mathcal{C}}$), and therefore by Theorem 3.1 the collection $\widetilde{\mathcal{L}}$ is generable in the limit with mistake feedback (in the noiseless model). Moreover, the generator $\mathcal{G}$ above is exactly the standard inner-cover generator (run on $\widetilde{\mathcal{C}}$ and avoiding $S_n$ and past outputs), so it generates from $\widetilde{K}$ in the noiseless sense. Therefore there exists $n^\star$ such that for all $n \geq n^\star$,

$$z_n \in \widetilde{K} \setminus S_n.$$

Finally, note that $\widetilde{K}$ differs from $K$ on the finite set $F := K \triangle \widetilde{K} \subseteq A \cup B$. Since the generator never repeats an output, each $f \in F$ can be output at most once. Hence there exists $n^\dagger$ such that for all $n \geq n^\dagger$ we have $z_n \notin F$. For all $n \geq \max(n^\star, n^\dagger)$, we have $z_n \in \widetilde{K}$ and $z_n \notin \widetilde{K} \triangle K$, which implies $z_n \in K$. Together with $z_n \notin S_n$, this yields that for any iteration $n$ with $n \geq \max(n^\star, n^\dagger)$,

$$z_n \in K \setminus S_n,$$

and, hence, the generator generates from $K$ in the limit under the eventually correct feedback transcript. Since the choice of $K \in \mathcal{L}$ and the interaction were arbitrary, the collection $\mathcal{L}$ is generable in the limit with eventually correct mistake feedback.

**Part 2 (If $\mathcal{L}$ is generable then it has a countable inner cover):** If $\mathcal{L}$ is generable in the limit with eventually correct mistake feedback, then in particular it is generable in the special case where the feedback is correct from the beginning (take $n_0 = 1$ in Definition 5). Thus, $\mathcal{L}$ is generable in the limit with (noiseless) mistake feedback, and by Theorem 3.1 it follows that $\mathcal{L}$ has a countable inner cover. $\qquad\square$

*Proof of Theorem A.7.* We divide the proof into two parts.

**Part 1 (Countable inner covers imply set-generation).** Assume that $\mathcal{L}$ has a countable inner cover $\mathcal{C} = \{C_1, C_2, \ldots\}$. We prove that $\mathcal{L}$ is set-generable in the limit with eventually correct query feedback.

**Step 1.1 (Finite expansion preserves countable inner covers):** We use the finite expansion subroutine of Mehrotra et al. (2025) to expand $\mathcal{L}$ into

$$\widetilde{\mathcal{L}} := \{ L \cup A \setminus B : L \in \mathcal{L},\ A \subseteq \mathcal{X},\ B \subseteq \mathcal{X},\ |A| < \infty,\ |B| < \infty \}\,.$$

In words, $\widetilde{\mathcal{L}}$ is obtained from $\mathcal{L}$ by adding and/or removing finitely many elements from each $L \in \mathcal{L}$. Similarly, define the finite expansion of the cover

$$\widetilde{\mathcal{C}} := \{ C_i \cup A \setminus B : i \in \mathbb{N},\ A \subseteq \mathcal{X},\ B \subseteq \mathcal{X},\ |A| < \infty,\ |B| < \infty \}\,.$$

As with $\widetilde{\mathcal{L}}$, $\widetilde{\mathcal{C}}$ is obtained from $\mathcal{C}$ by adding and/or removing finitely many elements from each $C_i$.

Since $\mathcal{X}$ is countable, the collection of finite subsets of $\mathcal{X}$ is countable, and hence $\widetilde{\mathcal{C}}$ is countable. Every set in $\widetilde{\mathcal{C}}$ is infinite, since it is obtained from an infinite $C_i$ by adding and removing finitely many elements. Moreover, the same argument as in Step 1.1 of the proof of Theorem A.6 shows that $\widetilde{\mathcal{C}}$ is a countable inner cover of $\widetilde{\mathcal{L}}$.

**Step 1.2 (Reduction to a noiseless expanded target, with non-repeating queries):** Fix an arbitrary target $K \in \mathcal{L}$ and consider an arbitrary interaction in the sense of Definition 2, except that the generator receives an eventually correct transcript $\widetilde{a}_1, \widetilde{a}_2, \ldots$ as in Definition 7. We construct a set-based generator that (i) ignores the adversary's examples as a source of information, (ii) never queries the same point twice, and (iii) reduces the contaminated transcript to an exact query-feedback transcript for a finite modification $K'$ of $K$.

Let $\prec$ be the canonical order on $\mathcal{X}$ induced by $(\overline{x}_1, \overline{x}_2, \ldots)$. Fix an enumeration $\widetilde{\mathcal{C}} = \{\widetilde{C}_1, \widetilde{C}_2, \ldots\}$. On round $n$, let

$$Q_n := \{y_1, \ldots, y_{n-1}\}$$

be the set of previously queried points, and note that $Q_n$ is finite. The generator also stores the observed label $\widetilde{a}_t$ together with the queried point $y_t$ for each $t < n$.

*Index selection (skipping witnessed bad candidates).* Define $i_n$ to be the smallest index $i \in \mathbb{N}$ such that there is *no* previously queried point $y_t \in \widetilde{C}_i$ with $\widetilde{a}_t = 0$, that is,

$$i_n := \min\{\, i \in \mathbb{N} : \forall t < n, \ \left( y_t \in \widetilde{C}_i \implies \widetilde{a}_t = 1 \right) \,\}.$$

Equivalently, the generator begins with $i = 1$ and repeatedly increments $i$ while it has already witnessed a queried point in $\widetilde{C}_i$ that received answer $0$.

*Choosing a fresh query point.* Let

$$T_n := Q_n \cap \widetilde{C}_{i_n}$$

be the set of points already queried that lie in the current candidate set $\widetilde{C}_{i_n}$. Choose the query point

$$y_n := \min_{\prec}\{x \in \widetilde{C}_{i_n} : x \notin Q_n\},$$

which is well-defined because $Q_n$ is finite and $\widetilde{C}_{i_n}$ is infinite. By construction, the query points never repeat: $y_n \notin Q_n$ for all $n$.

After receiving the membership answer $\widetilde{a}_n \in \{0, 1\}$, output the set

$$Z_n := \begin{cases} \widetilde{C}_{i_n} \setminus (S_n \cup Q_{n+1}) & \text{if } \widetilde{a}_n = 1, \\ \widetilde{C}_{i_n+1} \setminus (S_n \cup Q_{n+1}) & \text{if } \widetilde{a}_n = 0, \end{cases}$$

where $Q_{n+1} = Q_n \cup \{y_n\}$. Each $Z_n$ is infinite, hence the generator is set-based, and $Z_n \cap S_n = \emptyset$ by construction.

*Defining the expanded target $K'$.* By eventual correctness, there exists $n_0 \in \mathbb{N}$ such that for all $n \geq n_0$,

$$\widetilde{a}_n = \mathbb{1}\{y_n \in K\}.$$

Define finite sets

$$A := \{y_n : n < n_0, \ \widetilde{a}_n = 1, \ y_n \notin K\} \qquad \text{and} \qquad B := \{y_n : n < n_0, \ \widetilde{a}_n = 0, \ y_n \in K\}.$$

Both $A$ and $B$ are finite, as they are contained in $\{y_1, \ldots, y_{n_0-1}\}$. Let

$$K' := (K \cup A) \setminus B.$$

Then $K' \in \widetilde{\mathcal{L}}$ by definition of finite expansion, and moreover the transcript is *exactly consistent* with membership in $K'$ along the queried points: for every $n \in \mathbb{N}$,

$$\widetilde{a}_n = \mathbb{1}\{y_n \in K'\}.$$

Indeed, for $n < n_0$ this holds by the definition of $A$ and $B$, while for $n \geq n_0$ it holds because $y_n \notin \{y_1, \ldots, y_{n_0-1}\} \supseteq A \cup B$ (queries do not repeat), so $K'$ agrees with $K$ on $y_n$.

Thus, when viewed as an interaction with target language $K'$, the generator experiences a noiseless query-feedback transcript.

**Step 1.3 (Generate from $K'$ and conclude generation for $K$).** Since $\widetilde{\mathcal{L}}$ has a countable inner cover (Step 1.1), it is set-generable in the limit with (noiseless) query feedback by Theorem 3.4.

We now show directly that the above generator set-generates $K'$ in the noiseless query-feedback model. Recall from Step 1.2 that for every $n \in \mathbb{N}$,

$$\widetilde{a}_n = \mathbb{1}\{y_n \in K'\},$$

so we may view the interaction as one with target language $K'$ and correct query answers.

Since $\widetilde{\mathcal{C}}$ is a countable inner cover of $\widetilde{\mathcal{L}}$, there exists a least index $j^\star$ such that $\widetilde{C}_{j^\star} \subseteq K'$. We claim that there exists $n^\star$ such that for all $n \geq n^\star$,

$$Z_n \subseteq K' \setminus S_n.$$

First, consider any index $i < j^\star$. Since $\widetilde{C}_i \not\subseteq K'$, the set $\widetilde{C}_i \setminus K'$ is nonempty; let

$$u_i := \min_{\prec}\{x \in \widetilde{C}_i : x \notin K'\} \qquad \text{and} \qquad P_i := \{x \in \widetilde{C}_i : x \prec u_i\}.$$

As $\prec$ is induced by an enumeration of $\mathcal{X}$, the initial segment $\{x \in \mathcal{X} : x \prec u_i\}$ is finite, hence $P_i$ is finite. While the current index equals $i$, the generator queries the $\prec$-least element of $\widetilde{C}_i$ not in $Q_n$, and never repeats a query point. Therefore, after at most $|P_i| + 1$ rounds during which $i_n = i$, the generator queries $u_i$. Since $u_i \notin K'$, we have $\widetilde{a}_n = 0$ on that round, and hence $i$ becomes permanently invalidated: there now exists a previously queried point $y_t \in \widetilde{C}_i$ with $\widetilde{a}_t = 0$, so $i$ can never again satisfy the defining condition for $i_n$. Consequently, each index $i < j^\star$ can occur only finitely many times as $i_n$.

Since there are only finitely many indices less than $j^\star$, there exists a finite round $n^\star$ such that for all $n \geq n^\star$ we have $i_n \geq j^\star$. Moreover, the index $j^\star$ is never invalidated: if there were some $t$ with $y_t \in \widetilde{C}_{j^\star}$ and $\widetilde{a}_t = 0$, then $y_t \notin K'$, contradicting $\widetilde{C}_{j^\star} \subseteq K'$. Thus, for all sufficiently large $n$, the minimal non-invalidated index is exactly $j^\star$, and hence $i_n = j^\star$.

For every $n$ large enough that $i_n = j^\star$, we have $\widetilde{a}_n = 1$ because $y_n \in \widetilde{C}_{j^\star} \subseteq K'$, and therefore

$$Z_n = \widetilde{C}_{j^\star} \setminus (S_n \cup Q_{n+1}) \subseteq K' \setminus S_n \,.$$

This proves the claim.

We now conclude that eventually $Z_n \subseteq K \setminus S_n$. Fix any $n \geq \max(n^\star, n_0)$ and any $x \in Z_n$. Then $x \in K'$ and $x \notin S_n$. If $x \notin K$, then $x \in K' \setminus K \subseteq A \subseteq \{y_1, \ldots, y_{n_0-1}\} \subseteq Q_{n+1}$. However, by construction $Z_n$ is disjoint from $Q_{n+1}$, a contradiction. Hence $x \in K$, and therefore $x \in K \setminus S_n$. As $x \in Z_n$ was arbitrary, this shows that for all $n \geq \max(n^\star, n_0)$,

$$Z_n \subseteq K \setminus S_n,$$

as desired.

**Part 2 (Set-generation implies a countable inner cover).** If $\mathcal{L}$ is set-generable in the limit with eventually correct query feedback, then in particular it is set-generable in the special case where the feedback is correct from the beginning (take $n_0 = 1$ in Definition 7). Thus, $\mathcal{L}$ is set-generable in the limit with (noiseless) query feedback, and by Theorem 3.4 it follows that $\mathcal{L}$ has a countable inner cover. $\qquad\square$

### A.6. Proofs of Theorems 3.9 and 3.10 (Implications to Generation without Feedback)

In this section, we prove Theorems 3.9 and 3.10, which we restate below.

**Theorem 3.9** (Equivalence of set-based and element-based generators). *A collection $\mathcal{L}$ is generable in the limit by a set-based generator if and only if $\mathcal{L}$ is generable in the limit by an element-based generator.*

**Theorem 3.10** (Necessary and Sufficient Conditions). *The following statements hold for collections $\mathcal{L}$ in the no-feedback model.*

1. *If $\mathcal{L}$ has an inner-cover of finite size, then $\mathcal{L}$ is generable in the limit.*
2. *If $\mathcal{L}$ is generable in the limit, then $\mathcal{L}$ has a countable inner-cover (Definition 4).*

*Further, the following hold:*

1. *There is a collection $\mathcal{L}$ with a countable inner-cover that is not generable in the limit.*
2. *There is also a collection $\mathcal{L}$ that is generable in the limit and has a countable inner-cover.*
3. *There is also a collection $\mathcal{L}$ that is generable in the limit and does not have a finite inner-cover.*

### A.6.1. PROOF OF THEOREM 3.9

*Proof of Theorem 3.9.* We divide the proof into two parts corresponding to the two directions.

**Set-based $\Rightarrow$ element-based.** Let $G = (G_n)_{n \in \mathbb{N}}$ be a set-based generator that generates from every $K \in \mathcal{L}$ in the limit. Recall we have fixed a canonical ordering of $\mathcal{X}$, denoted $(\overline{x}_1, \overline{x}_2, \ldots)$. Define an element-based generator $G' = (G'_n)_{n \in \mathbb{N}}$ by

$$G'_n(h) := \overline{x}_i \quad \text{where} \quad i = \min\{i \in \mathbb{N} : \overline{x}_i \in G_n(h)\} \,.$$

In other words, $\mathcal{G}'$ outputs the smallest element of $\mathcal{G}_n(h)$ in the canonical ordering of the domain $\mathcal{X}$. This is well-defined because $\mathcal{G}_n(h)$ is infinite and hence nonempty. To see that $\mathcal{G}'$ is a valid generator, fix any $K \in \mathcal{L}$ and any enumeration of $K$, with ordered histories $H_n = (x_1, \ldots, x_n)$ and seen sets $S_n = \operatorname{set}(H_n)$. Since $\mathcal{G}$ generates from $K$ in the limit, there exists $n^\star$ such that for all $n \geq n^\star$,

$$\mathcal{G}_n(H_n) \subseteq K \setminus S_n \,.$$

For such $n$, we have $\mathcal{G}'_n(H_n) \in \mathcal{G}_n(H_n) \subseteq K \setminus S_n$. Hence $\mathcal{G}'$ generates from $K$ in the limit, and since $K \in \mathcal{L}$ was arbitrary, $\mathcal{L}$ is generable by an element-based generator.

**Element-based $\Rightarrow$ set-based.** Let $\mathcal{G} = (\mathcal{G}_n)_{n \in \mathbb{N}}$ be an element-based generator that generates $\mathcal{L}$ in the limit. We first replace it by the harmless freshened generator $\mathcal{G}^\circ$ defined by

$$\mathcal{G}_n^\circ(h) := \begin{cases} \mathcal{G}_n(h) & \text{if } \mathcal{G}_n(h) \notin \operatorname{set}(h) \,, \\ \min_\prec \{x \in \mathcal{X} : x \notin \operatorname{set}(h)\} & \text{otherwise.} \end{cases}$$

Since $\mathcal{G}$ eventually outputs elements of $K \setminus \operatorname{set}(H_n)$ on every enumeration of every $K \in \mathcal{L}$, the freshened generator agrees with $\mathcal{G}$ eventually on every successful run; hence $\mathcal{G}^\circ$ also generates $\mathcal{L}$ in the limit. Moreover, $\mathcal{G}_n^\circ(h) \notin \operatorname{set}(h)$ for every finite history $h$.

We will use the following sequence-version of the usual locking argument.

**Lemma A.8** (Self-locking prefixes for sequence generators). *Fix $K \in \mathcal{L}$, an enumeration $x_1, x_2, \ldots$ of $K$, and the corresponding histories $H_n = (x_1, \ldots, x_n)$. There exists a time $n_0 \in \mathbb{N}$ such that for every finite continuation $\rho = (v_1, \ldots, v_r) \in K^r$ and every $r \geq 0$,*

$$\mathcal{G}_{n_0+r}^\circ(H_{n_0} \parallel \rho) \in K \setminus \operatorname{set}(H_{n_0} \parallel \rho) \,.$$

*Proof of Lemma A.8.* Suppose otherwise. Then for every $n$ there is a finite continuation of the actual prefix $H_n$ over $K$ on which $\mathcal{G}^\circ$ makes a mistake at the end of the continuation. Using this failure condition, one constructs a counterexample enumeration by stages: at stage $i$, take a sufficiently long actual prefix $H_{n_i}$ containing the first $i$ elements of the fixed enumeration and extending all samples committed so far, then append a witnessing finite continuation on which the generator errs. The prefixes are chosen nested, and each stage forces one additional error while ensuring that $x_i$ appears. The limit sequence is therefore an enumeration of $K$ on which $\mathcal{G}^\circ$ makes infinitely many mistakes, contradicting that $\mathcal{G}^\circ$ generates from $K$ in the limit. Hence such an $n_0$ exists. $\square$

Now define a set-based generator $\widehat{\mathcal{G}} = (\widehat{\mathcal{G}}_n)_{n \in \mathbb{N}}$ as follows. On input an ordered history $h = (x_1, \ldots, x_n)$, recursively define

$$h^{(0)} := h, \qquad s_t := \mathcal{G}_{n+t-1}^\circ\left(h^{(t-1)}\right), \qquad h^{(t)} := h^{(t-1)} \parallel s_t \quad (t \geq 1),$$

and output

$$\widehat{\mathcal{G}}_n(h) := \{s_t : t \in \mathbb{N}\} \,.$$

Because $\mathcal{G}^\circ$ never outputs an element already in its input history, the strings $s_1, s_2, \ldots$ are pairwise distinct; hence $\widehat{\mathcal{G}}_n(h)$ is infinite for every input history $h$.

Fix a target $K \in \mathcal{L}$ and an enumeration of $K$. Let $n_0$ be the self-locking time from Lemma A.8. For any $n \geq n_0$, applying the lemma inductively to the continuation generated by $s_1, s_2, \ldots$ gives

$$s_t \in K \setminus \operatorname{set}\left(h^{(t-1)}\right) \qquad \text{for every } t \geq 1.$$

In particular, $s_t \in K \setminus S_n$ for every $t$, where $S_n = \operatorname{set}(H_n)$. Therefore

$$\widehat{\mathcal{G}}_n(H_n) \subseteq K \setminus S_n$$

for all sufficiently large $n$. Thus $\widehat{\mathcal{G}}$ set-generates $\mathcal{L}$ in the limit. $\square$

A.6.2. PROOF OF THEOREM 3.10 (NECESSARY AND SUFFICIENT CONDITIONS FOR GENERATION)

We divide Theorem 3.10, into the following five results.

**Theorem A.9.** *If a collection $\mathcal{L}$ has an inner cover of finite size, then $\mathcal{L}$ is generable in the limit (without feedback).*

**Theorem A.10.** *If $\mathcal{L}$ is generable in the limit (without feedback) by an element-based generator (equivalently, by a set-based generator), then $\mathcal{L}$ has a countable inner cover in the sense of Definition 4.*

**Theorem A.11.** *Generation in the limit (without feedback) is not characterized by the existence of a countable inner-cover.*

**Theorem A.12.** *There is a collection $\mathcal{L}$ which is generable in the limit and has a countable inner-cover.*

**Theorem A.13.** *There exists a language collection $\mathcal{L}'$ that is generable in the limit (without feedback) but does not admit any finite inner cover.*

In the remainder of this section, we prove the above theorems.

*Proof of Theorem A.9.* Assume $\mathcal{L}$ has a finite inner cover $\{C_1, \ldots, C_t\}$. For each $i \in \{1, \ldots, t\}$ define the subcollection $\mathcal{L}_i := \{L \in \mathcal{L} : C_i \subseteq L\}$. Then $\mathcal{L} = \bigcup_{i=1}^{t} \mathcal{L}_i$ by the cover property.

Fix $i \in \{1, \ldots, t\}$. Consider the set-based generator $\mathcal{G}^{(i)}$ defined on an ordered history $h_n$ by

$$\mathcal{G}_n^{(i)}(h_n) := C_i \setminus \mathrm{set}(h_n).$$

Since $C_i$ is infinite and $\mathrm{set}(h_n)$ is finite, $\mathcal{G}_n^{(i)}(h_n)$ is infinite for all $n$, so $\mathcal{G}^{(i)}$ is set-based. Moreover, for any target $K \in \mathcal{L}_i$ and any enumeration of $K$ with histories $H_n$ and seen sets $S_n = \mathrm{set}(H_n)$, we have $C_i \subseteq K$, hence for every $n$, $\mathcal{G}_n^{(i)}(H_n) = C_i \setminus S_n \subseteq K \setminus S_n$. Therefore $\mathcal{G}^{(i)}$ generates from every $K \in \mathcal{L}_i$ in the limit (indeed, from time $n^\star = 1$). In particular, each $\mathcal{L}_i$ is uniformly generable in the sense of (Li et al., 2025). By Li et al. (2025, Theorem 3.10), a finite union of uniformly generable collections is generable in the limit. Since $\mathcal{L} = \bigcup_{i=1}^{t} \mathcal{L}_i$, it follows that $\mathcal{L}$ is generable in the limit. $\square$

*Proof of Theorem A.10.* Assume that $\mathcal{L}$ is generable in the limit by an element-based generator. By Theorem 3.9, $\mathcal{L}$ is also generable in the limit by a set-based generator $\mathcal{G} = (\mathcal{G}_n)_{n \in \mathbb{N}}$. For each $n$, $\mathcal{G}_n$ takes as input an ordered history $h \in \mathcal{X}^n$. Since $\mathcal{X}$ is countable, $\mathcal{X}^n$ is countable, and hence the range $\mathrm{Range}(\mathcal{G}_n) := \{\mathcal{G}_n(h) : h \in \mathcal{X}^n\}$ is countable. Let $\mathcal{C} := \bigcup_{n \in \mathbb{N}} \mathrm{Range}(\mathcal{G}_n)$. Then $\mathcal{C}$ is a countable union of countable sets, hence countable. Enumerate its distinct elements as $\mathcal{C} = \{C_1, C_2, \ldots\}$, and let $\mathcal{C}_\infty \subseteq \mathcal{C}$ denote the subfamily consisting of the *infinite* sets among $\mathcal{C}$ (so $\mathcal{C}_\infty$ is still countable). We claim that $\mathcal{C}_\infty$ is a countable inner cover of $\mathcal{L}$. To see this, fix any target language $K \in \mathcal{L}$ and any enumeration $x_1, x_2, \ldots$ of $K$, with $h_n := (x_1, \ldots, x_n)$ and $S_n := \mathrm{set}(h_n)$. Let $Z_n := \mathcal{G}_n(h_n)$ be the output at time $n$. Since $\mathcal{G}$ generates from $K$ in the limit, there exists $n^\star$ such that for all $n \geq n^\star$,

$$Z_n \subseteq K \setminus S_n.$$

In particular, $Z_{n^\star} \subseteq K$. Moreover, since $\mathcal{G}$ is set-based, its outputs are languages, and in particular $Z_{n^\star}$ is infinite. Hence $Z_{n^\star} \in \mathcal{C}_\infty$, so there exists $C_i \in \mathcal{C}_\infty$ with $C_i = Z_{n^\star} \subseteq K$. As $K$ was arbitrary, $\mathcal{C}_\infty$ is a countable family of infinite sets such that every $K \in \mathcal{L}$ contains some $C_i \in \mathcal{C}_\infty$, which is exactly a countable inner cover. $\square$

*Proof of Theorem A.11.* Assume for contradiction that generation in the limit is characterized by countable inner-covers, that is, for every language collection $\mathcal{L} \subseteq 2^\mathcal{X}$, $\mathcal{L}$ is generable in the limit if and only if $\mathcal{L}$ has a countable inner-cover. Under this assumption, generation in the limit would be closed under finite unions: if $\mathcal{L}_1$ and $\mathcal{L}_2$ are each generable in the limit, then each has a countable inner-cover, and the union of these two covers is again countable, yielding a countable inner-cover for $\mathcal{L}_1 \cup \mathcal{L}_2$, and hence $\mathcal{L}_1 \cup \mathcal{L}_2$ would also be generable in the limit.

However, Hanneke et al. (2025) and Bai et al. (2026) construct two collections $\mathcal{L}_1, \mathcal{L}_2$ such that each of $\mathcal{L}_1$ and $\mathcal{L}_2$ is generable in the limit, but $\mathcal{L}_1 \cup \mathcal{L}_2$ is not generable in the limit. By Theorem A.10, each of $\mathcal{L}_1$ and $\mathcal{L}_2$ has a countable inner-cover, so the union of these two covers is a countable inner-cover for $\mathcal{L}_1 \cup \mathcal{L}_2$. Thus $\mathcal{L}_1 \cup \mathcal{L}_2$ is a concrete collection with a countable inner-cover that is not generable in the limit, and generation in the limit is not characterized by countable inner-covers. $\square$

*Proof of Theorem A.12.* It suffices to consider any countable collection $\mathcal{L}$: since all countable collections are generable (as proved by (Kleinberg & Mullainathan, 2024)) and since any countable collection is its own countable inner cover. $\square$

*Proof of Theorem A.13.* Let the domain be $\mathcal{X} \coloneqq \mathbb{N}^2$. For each $i \in \mathbb{N}$, define the (infinite) language $L_i \coloneqq \{(i,j) : j \in \mathbb{N}\}$. Notice that languages are pairwise disjoint and countably many. Let $\mathcal{L} \coloneqq \{L_i : i \in \mathbb{N}\}$.

**Part 1 ($\mathcal{L}$ is generable in the limit):** Consider the element-based generator $\mathcal{G}$ defined as follows. On input the ordered history $h_n = (x_1, \ldots, x_n)$, let $S_n = \mathrm{set}(h_n)$ and let $(i^\star, j^\star)$ be the first element $x_1$ of the history. The generator outputs $z_n \coloneqq \min\{(i^\star, j) : j \in \mathbb{N}, \ (i^\star, j) \notin S_n\}$, where the minimum is taken with respect to the canonical order on $\mathcal{X}$. This is well-defined because $S_n$ is finite and $L_{i^\star}$ is infinite.

Fix any target $K \in \mathcal{L}$ and any enumeration $x_1, x_2, \ldots$ of $K$. Then $K = L_{i^\star}$ for the unique $i^\star$ such that the first element satisfies $x_1 = (i^\star, j^\star)$ for some $j^\star$. Hence, for every $n \geq 1$, $z_n \in L_{i^\star} \setminus S_n = K \setminus S_n$. Thus, $\mathcal{G}$ generates from $K$ in the limit (indeed, from time $n^\star = 1$), and since $K$ was arbitrary, $\mathcal{L}$ is generable in the limit.

**Part 2 ($\mathcal{L}$ has no finite inner cover):** Assume for contradiction that $\{C_1, \ldots, C_t\}$ is a finite inner cover of $\mathcal{L}$. For each $r \in \{1, \ldots, t\}$, the set $C_r$ can be a subset of at most one of the pairwise-disjoint languages $L_i$, because $C_r$ is infinite. Let

$$I \coloneqq \{i \in \mathbb{N} : C_r \subseteq L_i \text{ for some } r \in \{1, \ldots, t\}\}$$

be the finite set of language indices covered by at least one of the finitely many cover elements. Choose $i^\dagger \in \mathbb{N} \setminus I$. Then, by definition of $I$, no $C_r$ is a subset of $L_{i^\dagger}$. This contradicts that $\{C_1, \ldots, C_t\}$ is an inner cover of $\mathcal{L}$, since it fails to cover $L_{i^\dagger}$. Therefore $\mathcal{L}$ admits no finite inner cover. $\square$

## A.7. Every Uniformly Generable Collection has a Countable Inner Cover

**Proposition A.14** (Uniform finite-time generation implies a countable inner cover). *Let $\mathcal{L}$ be a collection of languages over $\mathbb{N}$. Suppose there exist a deterministic* set-based *generator $\mathcal{G}$ and an integer $N \in \mathbb{N}$ with the following* uniform correctness-time *property:*

*for every target language $K \in \mathcal{L}$ and every run of $\mathcal{G}$ on an enumeration of $K$ (with whatever feedback transcript the model supplies), if $H_N$ denotes the ordered history at time $N$, then the generator's output at time $N$, denoted $\mathcal{G}_N(H_N, \beta_{1:N-1})$, is a language satisfying*

$$\mathcal{G}_N(H_N, \beta_{1:N-1}) \subseteq K.$$

*Then $\mathcal{L}$ admits a countable inner cover, meaning that there exists a countable family $\mathcal{C}$ of languages such that for every $K \in \mathcal{L}$ there exists $C \in \mathcal{C}$ with $C \subseteq K$.*

*Proof.* Consider the set of all length-$N$ histories that $\mathcal{G}$ can receive, namely pairs

$$h = (\sigma, \beta_{1:N-1})$$

where $\sigma \in \mathbb{N}^N$ is an ordered sample history and $\beta_{1:N-1}$ is a feedback sequence of length $N-1$ (over whatever feedback alphabet the model uses). Let

$$\mathcal{H}_N \coloneqq \left\{ (\sigma, \beta_{1:N-1}) : \sigma \in \mathbb{N}^N \text{ and } \beta_{1:N-1} \text{ is length } N-1 \right\}.$$

Since $\mathbb{N}$ is countable, the set $\mathbb{N}^N$ of ordered length-$N$ histories is countable. Also, the set of length-$N-1$ feedback sequences is countable (it is a finite Cartesian power of a countable set, or finite if the alphabet is finite). Therefore $\mathcal{H}_N$ is countable.

Define the family

$$\mathcal{C} \coloneqq \left\{ \mathcal{G}_N(h) : h \in \mathcal{H}_N \right\}.$$

Because $\mathcal{H}_N$ is countable and $\mathcal{G}$ is deterministic, $\mathcal{C}$ is countable.

It remains to show that $\mathcal{C}$ is an inner cover of $\mathcal{L}$. Fix any $K \in \mathcal{L}$ and consider any run of $\mathcal{G}$ on an enumeration of $K$. Let $h^\star = (H_N, \beta_{1:N-1})$ be the length-$N$ ordered history observed in that run. By definition, $h^\star \in \mathcal{H}_N$, so $C^\star := \mathcal{G}_N(h^\star) \in \mathcal{C}$. By the uniform correctness-time assumption,

$$C^\star \;=\; \mathcal{G}_N(H_N, \beta_{1:N-1}) \;\subseteq\; K.$$

Thus every $K \in \mathcal{L}$ contains some member of $\mathcal{C}$, and $\mathcal{C}$ is a countable inner cover of $\mathcal{L}$. $\qquad\square$

