# OpenReview forum: "Language Generation with Feedback: Queries and Mistakes"
_ICML.cc/2026/Conference — ICML 2026 regular_

### Official Review · Reviewer_e5eK · 2026-03-10

**Soundness:** 2
**Presentation:** 2
**Significance:** 2
**Originality:** 2
**Overall Recommendation:** 3
**Confidence:** 1

**Summary:**

The paper investigates langauge generation in the limit with two variants of actions: mistake feedback and the ability to query the membership of the string to the target langauge. The main result provides a characterization of collections of langauge that are generable with mistake feedback, extending to set-based generators. The implication of the results is a new closure properties for generation using unions.

**Compliance With Llm Reviewing Policy:**

Affirmed.

**Key Questions For Authors:**

- What is the difference between getting the feedback of mistakes, and querying whether the string belongs to the language or not. When the feedback says that a mistake has been made (variant 1), doesn't it imply that the string does not belong to the target language?

- How valid is the assumption of having countable inner-cover?

**Strengths And Weaknesses:**

The paper is theoretically rich, where the characterisation of what is learnable vs not is determined by having inner-covers -- a property of the language of possessing implicit structure. Authors make a differentiation between element- and set-based generators, which is an interesting perspective of the capability of the generator. The two action variants appear related with subtle distinction, which becomes important when distinguishing between element-based and set-based generators.

Since the paper focuses on theory, there is no practical experiments. The additional actions make the paper an incremental one.

---

> ### Author Rebuttal · Authors · 2026-03-31
>
> Thank you for taking the time to review our paper. We are happy that you found it theoretically rich.
>
> **Since the paper focuses on theory, there is no practical experiments.**
>
> Regarding the theoretical nature of our paper, we would like to point out that language generation in the limit is a rapidly growing area in learning theory and several papers in this line of work have been published at NeurIPS/ICML/ICLR without experimental material, including Kleinberg and Mullainathan [NeurIPS’24], Raman and Raman [ICML’25], Peale, Raman, and Reingold [ICML’25], and Hanneke, Karbasi, Mehrotra, and Velegkas [NeurIPS’25]. Thus, we do not view this aspect of our work as grounds for rejection, especially since, as the reviewer pointed out, the results are theoretically rich.
>
> **The additional actions make the paper an incremental one.**
>
> We are a bit confused by this, as you mentioned our results are “theoretically rich” and have theoretical implications that were not known in prior work, so we respectfully disagree that our results are incremental.
>
> **What is the difference between getting the feedback of mistakes, and querying whether the string belongs to the language or not. When the feedback says that a mistake has been made (variant 1), doesn't it imply that the string does not belong to the target language?**
>
> This is a great point: Indeed, mistake feedback and query feedback look very similar, but there is a subtle yet crucial difference. When one has query feedback and they want to find a label of a given point, they can query for its label prior to outputting it; if the label is 1 (meaning it is part of the target) they can safely output it, and if the label is 0 (meaning it is outside of the target) they can ignore it. However, in the case of mistake feedback, in order to find out the label of the point the generator has to output it; if the label ends up being 0, the generator will incur a mistake and incur a loss in this round. This is why element-based generation with mistake feedback does not admit the same characterization as element-based generation with query feedback (Theorem 3.3).
>
> **How valid is the assumption of having countable inner-cover?**
>
> The countable inner-cover condition is quite permissive: For instance, it holds for all countable language collections (like the ones recognized by DFAs or NFAs). Furthermore, this condition extends to many uncountable collections, such as all languages containing an infinite arithmetic progression. Our results in this paper allow us to understand precisely what is generable under this type of feedback when we deal with *uncountable* collections of languages.
>
> We extend the results to the uncountable setting since, typically, in learning theory, we wish to build a complete understanding of what is learnable without restricting to countable objects. Moreover, we would like to highlight that the countable inner-cover condition is not an assumption we make for our results to go through; rather, it is a necessary and sufficient condition for generation under the types of feedback we discuss in our work.
>
> **Regarding the soundness and presentation scores:**
>
> We are a bit confused by the soundness and presentation scores of 2, which usually indicate technical flaws or significant presentation issues. If the reviewer has concerns about any of our results or suggestions for improving the presentation, we would be glad to address them.

---

> > ### Author Rebuttal · Reviewer_e5eK · 2026-04-01
> >
> > Thanks for the rebuttal.
> >
> > - Many prior works may ignore experimental evaluations and yet are accepted in premier conferences. I am not certain how many years in the future this line of argument would sustain.
> >
> > - The paper is incremental, because it is built on top of many prior works.
> >
> > - The paper may be theoretically sound, but in terms of implementation, there is no evidence. The authors themselves agree on that.

---

> > > ### Author Response · Authors · 2026-04-02
> > >
> > > Thank you for your quick response. We are sorry to see that you reduced the overall score following our rebuttal.
> > >
> > > We address the three points raised below.
> > >
> > > **On the absence of experimental evaluation.** Theory of machine learning is explicitly listed among the main topics in the ICML 2026 call for papers, and purely theoretical papers are regularly accepted and recognized at leading ML venues. For instance, [CHMS'25] was a runner-up for the best paper award at NeurIPS 2025, without any experimental evaluation. We positioned our submission transparently under **Theory → Learning Theory** so that its nature and intended evaluation criteria would be clear.
> > >
> > > **On the claim that the paper is "incremental, because it is built on top of many prior works."** We respectfully disagree. The reviewer states that "the paper is incremental, because it is built on top of many prior works." Building on prior literature is a normal and essential part of scientific progress, not evidence of incrementality. The relevant question is whether the paper contributes new ideas and results. As the reviewer acknowledged in the original review, the paper is "theoretically rich." The paper introduces inner-covers as a new structural condition, and provides characterizations that were not known in prior work.
> > >
> > > **On the absence of implementation evidence.** We did not present the paper as an empirical contribution. For a theoretical paper, validity is established through rigorous definitions, results, and proofs. The ICML 2026 reviewing guidelines state that soundness may be supported by "theoretical analysis or experimental results," depending on the nature of the submission.
> > >
> > > We hope this clarifies our position. We are happy to address any further technical questions you may have.
> > >
> > > **Reference:** Optimal Mistake Bounds for Transductive Online Learning, Zachary Chase, Steve Hanneke, Shay Moran, Jonathan Shafer, NeurIPS 2025.

---

### Official Review · Reviewer_Z9pR · 2026-03-12

**Soundness:** 4
**Presentation:** 4
**Significance:** 3
**Originality:** 3
**Overall Recommendation:** 5
**Confidence:** 3

**Summary:**

The paper considers the model of language generation in the limit introduced by Kleinberg and Mullainathan (2024). In this setting, an adversary chooses a target language $K$ from a collection $\mathcal{L}$ and reveals positive examples sequentially; after observing the examples seen so far, the learner must output new strings. The learner succeeds if from after some finite time step it always outputs previously unseen strings that belong to $K$.

The authors study how this problem changes when the learner receives feedback at each step. They consider two feedback models: mistake feedback, where the learner is told whether its latest output was correct, and query feedback, where the learner can ask whether a chosen string belongs to the target language. The main result is a combinatorial characterization of generability in terms of "countable inner-covers": a class is generable with mistake feedback iff it has a countable inner-cover, and the same characterization holds for set-based generation with query feedback. Conceptually, the result is in the vein of learning-theoretic characterizations such as VC dimension or Littlestone dimension. The paper also derives several consequences and further results in this model, notably establishing robustness to corrupted feedback.

**Compliance With Llm Reviewing Policy:**

Affirmed.

**Final Justification:**

I'm happy to maintain my original score.

**Key Questions For Authors:**

The paper motivates the feedback model partly by analogy to modern LLM training pipelines (e.g. Remark 3.5). However, in current LLM systems, feedback is typically noisy, indirect, or comparative rather than a binary correctness signal. Do you expect the main characterization results to extend to models with noisy or preference-based feedback, or would such settings likely require fundamentally different techniques?

**Limitations:**

Yes

**Strengths And Weaknesses:**

Overall this is a nice paper. The line of work on language generation in the limit is quite interesting, and incorporating feedback is a very well-motivated and important direction. A clean combinatorial characterization such as the one in Theorem 3.1 is quite satisfying. The paper is clearly written; even without reading the proofs in detail, the proof sketches and overall structure are easy to follow.

The main limitation is that the model is quite stylized and abstract, so the connection to empirical LLM systems is currently fairly distant. However, this is not a major weakness given that the contribution is primarily theoretical. The paper also leaves some open questions (such as, the characterization of element-based query generation)but these seem like natural directions for future work rather than shortcomings of the current paper.

---

> ### Author Rebuttal · Authors · 2026-03-31
>
> Thank you for taking the time to read our paper. We are happy to see that you liked our work.
>
> **The paper motivates the feedback model partly by analogy to modern LLM training pipelines (e.g. Remark 3.5). However, in current LLM systems, feedback is typically noisy, indirect, or comparative rather than a binary correctness signal. Do you expect the main characterization results to extend to models with noisy or preference-based feedback, or would such settings likely require fundamentally different techniques?**
>
> That is a great question. We agree that feedback can be noisy in practice. In Section 3.4.3, we explain how noisy feedback can be handled: Our characterization remains robust to any finite amount of noise in the feedback that we obtain and also robust to *arbitrary* noise in the examples shown by the adversary.
>
> Regarding the preference-based feedback, in the current language generation model the output has a binary score (correct or incorrect), so it is not clear to us what preference-based feedback could look like. One can envision a model where different responses have different scores, and preference-based feedback would indicate which response achieves a better score. We believe that our ideas will be useful in that model, but we cannot claim with certainty that the results will generalize immediately, since it is not even clear yet what the appropriate generalization of the model is, though we view this as a very interesting direction for future work.

---

> > ### Author Rebuttal · Reviewer_Z9pR · 2026-04-03
> >
> > Thanks for the rebuttal. I maintain my positive review.

---

### Official Review · Reviewer_oYe7 · 2026-03-13

**Soundness:** 2
**Presentation:** 3
**Significance:** 3
**Originality:** 3
**Overall Recommendation:** 4
**Confidence:** 2

**Summary:**

This paper focuses on language generation in the limit when the generator receives feedback.
Two kinds of feedbacks are considered, i.e.,  mistake feedback (whether the last output was correct) and query feedback (membership queries for the target language).
Overall, the authors investigate a central theme: which language collections are generable under these feedback models. The setting builds on Kleinberg & Mullainathan (2024) and Charikar & Pabbaraju (2025a): an adversary chooses a target language $K$ from a collection $\mathcal{L}$ and enumerates examples; the generator outputs strings (element-based: one string per round; set-based: an infinite set per round) and receives feedback. The main results are characterizations: generability with mistake feedback is equivalent to the collection having a countable inner-cover (Definition 4); with query feedback, set-based generability also coincides with countable inner-cover, while element-based and set-based query feedback are not equivalent (Theorem 3.3). The authors derive closure properties (e.g., countable unions), generation with zero examples, robustness to contamination in the stream and in the feedback, and new implications for the no-feedback setting (e.g., generability “sandwiched” between finite and countable inner-covers). Proofs use tell-tale sets, simulation arguments, and inner-cover constructions.

**Compliance With Llm Reviewing Policy:**

Affirmed.

**Key Questions For Authors:**

I have two questions regarding the overall setup described in the paper.

1. Could the authors clarify what is meant by “generation *in the limit*”? How does it differ from ordinary (finite-time) generation, and what are its practical applications?

2. Could the authors provide a concrete example illustrating *generation with feedback*? For instance, it would be helpful to see a step-by-step example involving a simple language collection, where a generator operates using either mistake feedback or query feedback. Example 2.1 does not appear to have feedback.

**Limitations:**

I cannot find the discussion on limitations in this paper.

**Strengths And Weaknesses:**

**Strengths**

1. **Originality and significance.** The work gives the first characterizations of generability with mistake feedback and (for set-based) with query feedback, unifying them via the countable inner-cover notion. The equivalence of element-based and set-based under mistake feedback (Theorem 3.2) and the separation under query feedback (Theorem 3.3) are non-trivial. The implications (closure, zero-example generation, contamination robustness, and no-feedback implications) add substantial value. Related work (Kleinberg & Mullainathan, Bai et al., Charikar & Pabbaraju, Mehrotra et al., etc.) is discussed and positioned clearly.

2. **Theoretical soundness.** Definitions (generation in the limit, mistake/query feedback, inner-cover) are precise. The characterizations (Theorems 3.1–3.4) are stated clearly and proofs are delegated to the appendix. The technical overview (tell-tales, simulation, inner-cover connections) helps the reader follow the proof structure. The ratio of definitions to results is reasonable; the inner-cover is a compact condition that yields multiple consequences.

3. **Presentation.** The narrative is clear: model and preliminaries, then results (inner-cover, mistake feedback, query feedback), then implications. Examples (e.g., length-threshold languages, arithmetic progressions) aid intuition. The connection to LLM training (Remark 3.5) is a useful bridge to practice.

**Weaknesses**

My main concern is that the paper is purely theoretical; there are no experiments or simulations.
For a theory paper this is acceptable, but the reader may wonder how restrictive the countable inner-cover condition is for natural language collections (e.g., context-free or regular languages) and whether the feedback models are realistic for LLM training (e.g., mistake feedback as preference or reward signal). A short discussion or example mapping real-world feedback to the mistake/query models would strengthen the practical relevance.

---

> ### Author Rebuttal · Authors · 2026-03-31
>
> We would like to thank the reviewer for taking the time to read our paper. We are happy that you found our results non-trivial and the presentation clear.
>
> **My main concern is that the paper is purely theoretical.**
>
> Regarding the theoretical nature of our paper, we would like to point out that language generation in the limit is a rapidly growing area in learning theory and several papers in this line of work have been published at NeurIPS/ICML/ICLR without experimental material, including Kleinberg and Mullainathan [NeurIPS’24], Raman and Raman [ICML’25], Peale, Raman, and Reingold [ICML’25], and Hanneke, Karbasi, Mehrotra, and Velegkas [NeurIPS’25]. Thus, we do not view this aspect of our work as grounds for rejection, especially since, as the reviewer pointed out, the results are non-trivial and have interesting implications.
>
> **How restrictive the countable inner-cover condition is for natural language collections (e.g., context-free or regular languages).**
>
> The notion of countable inner-cover is not restrictive for several natural language collections such as regular and context-free grammars. Indeed, both of these collections are countable, so they immediately admit a countable inner cover. In particular, in the context of exact language learning, these types of queries for DFAs have been explored in the past and have led to celebrated algorithms, like Angluin’s $L^*$ algorithm which requires *both* membership queries and mistake feedback. Our results in this paper allow us to understand what is generable under this type of feedback when we deal with *uncountable* collections of languages. We extend the results to the uncountable setting since, typically, in learning theory, we wish to build a complete understanding of what is learnable without restricting to countable objects. Moreover, we would like to highlight that the countable inner-cover condition is not an assumption we make for our results to go through; rather, it is a necessary and sufficient condition for generation under the types of feedback we discuss in our work.
>
> **A short discussion or example mapping real-world feedback to the mistake/query models would strengthen the practical relevance.**
>
> We can make the following (abstract) connection: we can think of membership queries as the LLM asking the user for clarifications when there is ambiguity in the responses it can give, and we view mistake feedback as the user providing information to the LLM after the user sees a response the LLM generated (which happens in LLMs, in the form of having an (optional) thumbs up / down button in the response). We would be happy to include and expand upon this discussion in the updated version of our paper.
>
> **Could the authors clarify what is meant by “generation in the limit”? How does it differ from ordinary (finite-time) generation, and what are its practical applications?**
>
> Generation in the limit means that the learner stops outputting elements that don't belong to the target language $K$ after a finite number of timesteps, but we may not be able to provide a bound on how long it will take until the generator stops making mistakes. For instance, for some languages this may require 10 timesteps, while for other languages in the collection it might need 1000 timesteps; such a finite number though always exists. This is the precise definition that has been used throughout the line of work on language generation in the limit (and earlier work on language identification in the limit).
>
> **Could the authors provide a concrete example illustrating generation with feedback?**
>
> Thanks for the suggestion. Consider this concrete example. Let $\mathcal{X} = \mathbb{N}$ and partition it into blocks $B_j = \\{3j, 3j+1, 3j+2\\}$. Consider the collection $\mathcal{L}$ where each target language $K \in \mathcal{L}$ contains exactly one of two patterns for every $j \in \mathbb{N}$: either $\\{3j, 3j+1\\}$ or $\\{3j+2\\}$. Suppose the adversary selects the enumeration $x_1, x_2, \dots$. After observing $S_n$, the generator needs to output a valid, unseen string. Without feedback, guessing the pattern for a completely unseen block $B_j$ could easily result in a mistake. With query feedback, a valid element-based generator simply finds the first block $B_{j_n}$ disjoint from $S_n$ and queries whether $3j_n \in K$. It receives the binary answer $a_n$. If $a_n = 1$, it safely outputs $3j_n+1$; if $a_n = 0$, it safely outputs $3j_n+2$. We will add this example, as well as an example showing how mistake feedback works, in the next version of our work.
>
> **Regarding the soundness score:**
>
> We were also glad the reviewer found our proof overviews helpful to understand the structure of the proofs. That said, we were a bit confused with the soundness score of 2, which usually indicates technical flaws in a paper. If the reviewer is concerned about any of our results, we would be happy to clarify.

---

### Official Review · Reviewer_1row · 2026-03-19

**Soundness:** 3
**Presentation:** 3
**Significance:** 3
**Originality:** 3
**Overall Recommendation:** 4
**Confidence:** 1

**Summary:**

*I do not think I am the right audience for this paper, as I have no background in learning theory.*

Based on my rough understanding, this paper builds on a prior theoretical framework, “language generation in the limit,” and extends it by incorporating two types of feedback: query feedback and mistake feedback. The authors introduce the concept of a countable inner-cover and show that it provides a clear if-and-only-if characterization of generability under both feedback settings. The paper also derives several implications, including closure properties, robustness to noise, and generation without examples.

**Compliance With Llm Reviewing Policy:**

Affirmed.

**Final Justification:**

I maintain a low confidence rating due to my limited background in this area.

**Key Questions For Authors:**

* It seems that there are several abstract connections between the concepts in this paper and practical algorithms used in LLM training. For example, the notion of feedback in this paper appears conceptually similar to rewards or human feedback in RL-based methods. Is there any concrete mapping between your theory and modern LLM training practices? For instance, what would correspond to an “inner-cover” in real data distributions?

**Limitations:**

yes

**Strengths And Weaknesses:**

*As mentioned above, I do not think I am able to provide technical comments.*

That said, the paper is overall well written, which gives me some sense of the high-level structure of the theoretical framework.

---

> ### Author Rebuttal · Authors · 2026-03-31
>
> Thank you for supporting our paper. We are grateful for the time and effort you have spent on reviewing. We answer your question below.
>
> **It seems that there are several abstract connections between the concepts in this paper and practical algorithms used in LLM training. For example, the notion of feedback in this paper appears conceptually similar to rewards or human feedback in RL-based methods. Is there any concrete mapping between your theory and modern LLM training practices? For instance, what would correspond to an “inner-cover” in real data distributions?**
>
> We agree that there are several abstract connections between the concepts in our work and practical algorithms for LLM training; we mostly view our framework as a theoretical abstraction that allows us to study empirical phenomena in LLMs from a theoretical lens and obtain some insights. For instance, our framework shows that using feedback in the training process (which is akin to rewards in LLM post-training), allows us to bypass several lower bounds that hold in the absence of feedback.
>
> That said, it is not yet clear what the direct connections are between our learning primitives and LLM training. A potential interpretation of the inner-cover that you mentioned could be that if one is trying to learn a complicated set of distributions $\mathcal{D}$, some strategy might be to obtain a different (and simpler) set of distributions $\mathcal{D}'$ such that for every distribution $D \in \mathcal{D}$ there is some distribution $D' \in \mathcal{D}'$ that is "non-hallucinating" with respect to $D$; then, instead of learning the complicated distributions in $\mathcal{D}$, one can reduce to the simpler task of learning simpler distributions in $\mathcal{D}'.$ We hope that communicating these results to practitioners can lead to fruitful discussions that can bridge the gap between the theoretical model and practical LLM training.

---

> > ### Author Rebuttal · Reviewer_1row · 2026-04-06
> >
> > Thank you for your thoughtful response to my question!

---

### Decision · Program_Chairs · 2026-04-30

**Decision:**

Accept (regular)

**Comment:**

For this paper, the reviewer with the highest confidence has given a strongly positive evaluation, and the rationale behind that judgment appears convincing. On the other hand, some reviewers have concerns about the lack of experimental evaluation. In response, the authors have argued that this is customary in this research area.

In my view, the concern raised by those reviewers is reasonable. Even if omitting experimental evaluation is standard practice, it is important to consider diverse approaches to evaluation, and explaining this would be valuable for the ICML community. That said, given the developmental trajectory of this research area, even a theory-focused paper without experimental evaluation can be recognized as having a potential audience and future impact. Overall, the evaluations appear to lean positive.

Therefore, I recommend acceptance of this paper. However, I strongly recommend that the authors reflect the reviewers' comments regarding experimental evaluation in the revised version.